# Structural basis of GSK-3 inhibition by N-terminal phosphorylation and by the Wnt receptor LRP6

Jennifer L Stamos[1,2], Matthew Ling-Hon Chu[1,2], Michael D Enos[1,2], Niket Shah[1,2], William I Weis[1,2]*

[1]Department of Structural Biology, Stanford University, Stanford, United States; [2]Department of Molecular and Cellular Physiology, Stanford University, Stanford, United States

**Abstract** Glycogen synthase kinase-3 (GSK-3) is a key regulator of many cellular signaling pathways. Unlike most kinases, GSK-3 is controlled by inhibition rather than by specific activation. In the insulin and several other signaling pathways, phosphorylation of a serine present in a conserved sequence near the amino terminus of GSK-3 generates an auto-inhibitory peptide. In contrast, Wnt/β-catenin signal transduction requires phosphorylation of Ser/Pro rich sequences present in the Wnt co-receptors LRP5/6, and these motifs inhibit GSK-3 activity. We present crystal structures of GSK-3 bound to its phosphorylated N-terminus and to two of the phosphorylated LRP6 motifs. A conserved loop unique to GSK-3 undergoes a dramatic conformational change that clamps the bound pseudo-substrate peptides, and reveals the mechanism of primed substrate recognition. The structures rationalize target sequence preferences and suggest avenues for the design of inhibitors selective for a subset of pathways regulated by GSK-3.

*For correspondence: bill.weis@stanford.edu

**Competing interests:** The authors declare that no competing interests exist.

**Reviewing editor**: Philip Cole, Johns Hopkins University School of Medicine, United States

## Introduction

Glycogen synthase kinase-3 (GSK-3) is a highly conserved kinase (*Ali et al., 2001*) that in higher animals functions in insulin signaling, microtubule regulation, inflammatory pathways, and developmental programs including Hedgehog, Notch and Wnt signaling (*Jope and Johnson, 2004*; *Kaidanovich-Beilin and Woodgett, 2011*). GSK-3 has two isoforms in vertebrates, α and β, that share 97% identity in their catalytic cores but have different N- and C-terminal extensions. GSK-3 prefers pre-phosphorylated substrates, termed 'primed' substrates, with a phosphorylated serine or threonine residue at the position 4 residues C-terminal (P+4) to the serine or threonine to be phosphorylated (P+0) (*Fiol et al., 1987*). Over 70 P+4 primed substrates for GSK-3 have been identified and confirmed as true biological GSK-3 targets (*Sutherland, 2011*). GSK-3 is a potential therapeutic target for diabetes and neurological disorders (*Amar et al., 2011*), but its diversity of substrates complicates efforts to target a specific function.

The overall structure of the GSK-3 catalytic domain is similar to other protein kinases, containing an N-terminal lobe composed primarily of β-strands in a β barrel conformation, and a C-terminal lobe composed mostly of α-helices (*Dajani et al., 2001*; *ter Haar et al., 2001*). At the interface of the N- and C- lobes, the C-helix and activation loop of GSK-3 form the putative substrate-binding site and help to position residues involved in the binding of ATP and substrate catalysis (*Dajani et al., 2001*; *ter Haar et al., 2001*). Crystal structures of GSK-3 obtained in the absence of nucleotide and/or peptide substrate reveal an ordered activation loop similar in conformation to that found in other kinases when the loop is phosphorylated. GSK-3 also shows no evidence for movements of the C-helix critical for positioning of catalytically important residues (*Jura et al., 2011*). GSK-3 can

**eLife digest** Cells need to be able to respond to changes in the body, such as changes in hormone levels or the arrival of a pathogen such as a virus. Proteins acting in signaling pathways—where one protein switches 'on' or 'off' the next protein in the pathway—allow the detection of different changes or signals to be translated in the appropriate response.

The properties of a protein often depend on its shape, and many proteins change shape when they are switched 'on' and 'off'. Moreover, the ability to change shape allows a protein to interact with many other proteins and to be involved in many different signaling pathways.

The enzyme GSK-3 is a protein that is involved in several pathways, and it controls other proteins by adding chemical tags, called phosphate groups, to them. Unlike many other enzymes, GSK-3's default state is to be permanently switched on, and it can be switched off in a number of different ways. When a cell detects the hormone insulin, for example, another enzymes adds a phosphate group to a site near one end of GSK-3 to switch it off. Alternatively, when a cell recognises a different signaling molecule, called Wnt, a phosphate group is added to yet another protein, which then binds to and switches off GSK-3.

To explore the workings of GSK-3 in greater detail, Stamos et al. solved the three-dimensional structure of the enzyme that had been switched off in these two ways. In the insulin pathway, the region near to one end of GSK-3 that contains the added phosphate group was shown to bind to and block the site on the enzyme that usually binds to its target (i.e., to the next protein in the signaling pathway). In the Wnt pathway, remarkably, the same site on GSK-3 was blocked in a very similar way by the piece of the other protein with the phosphate group added.

GSK-3 is a potential drug target for the treatment of several diseases, such as diabetes and neurological disorders. However, as this enzyme is involved in multiple pathways, it has been hard to find drugs to treat any one condition without side effects. Uncovering subtle differences in how GSK-3 can be controlled in different pathways could, in the future, help with efforts to develop more specific drugs to target GSK-3 to treat these diseases.

be phosphorylated at a tyrosine (Tyr216) on its activation loop, which increases enzymatic activity approximately fivefold, a modest increase compared to the activation of other kinases (***Dajani et al., 2003***).

GSK-3 differs from most kinases in that it is constitutively active and controlled by inhibition, rather than by specific activation conferred by phosphorylation of the activation loop. Phosphorylation of a serine present in a conserved sequence near the amino terminus of the protein (Ser9 in GSK-3β) is the major regulatory checkpoint in most pathways, including insulin signaling (***Kaidanovich-Beilin and Woodgett, 2011***). For example, in the absence of Ser9 phosphorylation, GSK-3 phosphorylates glycogen synthase and renders it less active, and inactivates eIF2B. Binding of insulin to its cell surface receptor results in the activation of the kinase AKT/PKB, which phosphorylates the conserved N-terminal peptide, resulting in increased glycogen and protein synthesis (***Cross et al., 1995***; ***Frame and Cohen, 2001***). The phosphorylated sequence, designated here as the pS9 peptide, is thought to auto-inhibit GSK-3 by acting as a pseudo-substrate that blocks binding of other substrates (***Dajani et al., 2001***; ***Frame et al., 2001***).

GSK-3 also plays a critical role in the Wnt/β-catenin pathway. Binding of a Wnt growth factor to its cell surface receptors Frizzled and LRP5/6 results in the stabilization of the transcriptional co-activator β-catenin and activation of target genes. In the absence of Wnt, β-catenin is found in a 'destruction complex' that includes GSK-3, casein kinase 1 (CK1), the Adenomatous Polyposis Coli protein, and Axin. Axin has binding sites for β-catenin, GSK-3 and CK1, and thereby scaffolds efficient phosphorylation of the N-terminus of β-catenin, which marks β-catenin for ubiquitylation and proteasomal degradation (***Stamos and Weis, 2013***). The association with Axin sequesters a fraction of cytosolic GSK-3 in the destruction complex, which likely prevents interactions with regulatory proteins not involved in Wnt signaling, including kinases that target Ser9. Mutation of GSK-3β Ser9 does not affect Wnt signaling, and this residue does not become phosphorylated during Wnt stimulation (***Ding et al., 2000***; ***McManus et al., 2005***). Although isoform-specific roles have been observed in other pathways

(e.g., [*McManus et al., 2005*; *Patel et al., 2008*; *Kaidanovich-Beilin et al., 2009*]), GSK-3α and β function redundantly in Wnt signaling (*Doble et al., 2007*).

Upon Wnt activation, Axin-bound GSK-3 translocates to the plasma membrane, where phosphorylated peptide repeat motifs in the cytoplasmic tail of the activated LRP5/6 receptor interact with the β-catenin destruction complex to inhibit its activity (*MacDonald and He, 2012*). The five repeats, designated a-e, have the consensus sequence P-P-P-S/T-P-X-S/T, where each of the S/T residues becomes phosphorylated in response to Wnt (*Niehrs and Shen, 2010*). Several kinases, including GSK-3 itself, may be responsible for phosphorylating the first Ser/Thr residue of the motif (*Zeng et al., 2005*; *Chen et al., 2009*; *Cervenka et al., 2011*). Phosphorylation of this site forms a primed substrate recognition sequence for CK1, which then phosphorylates the second Ser/Thr residue (*Davidson et al., 2005*). Current models suggest that the phospho-LRP6 motifs can directly inhibit GSK-3 phosphorylation of β-catenin by engaging the kinase as a pseudo-substrate (*Piao et al., 2008*; *Wu et al., 2009*; *Kim et al., 2013a*). Although both phosphorylation sites in the consensus motif appear to be required for biological activity, only the first site is necessary for LRP6 to inhibit GSK-3 enzymatic activity (*Piao et al., 2008*). Additionally, the five repeats function cooperatively in LRP6, as removing any single motif from LRP6 results in significant decreases in Wnt signaling capability (*MacDonald et al., 2008*).

A major challenge has been to understand how GSK-3 is controlled such that its effects in one pathway are insulated from others. Some specificity is achieved through association with pathway-specific scaffolding proteins that bind GSK-3 and its substrate, such as the β-catenin destruction complex, which sequester GSK-3 from other pathways and lead to highly efficient phosphorylation. However, the inhibition by the LRP5/6 cytoplasmic domain demonstrates that other modes of inhibition can operate. We set out to assess the mechanistic similarities and differences between inhibition by N-terminal phosphorylation and inhibition mediated by the phosphorylated LRP5/6 motifs.

There are only a handful of crystal structures of kinases bound to peptide substrates (*Madhusudan et al., 1994*; *Hubbard, 1997*; *Lowe et al., 1997*; *Brown et al., 1999*; *Lippa et al., 2008*; *Lopez-Ramos et al., 2010*; *Soundararajan et al., 2013*), and only one, a tyrosine kinase, with a primed substrate (*Davis et al., 2009*). Here we show that the auto-inhibitory pS9 peptide sequence and the inhibitory LRP6 c- and e-motifs all bind to GSK-3 as pseudo-substrates. Unique to GSK-3, binding is associated with a drastic conformational rearrangement of a highly conserved loop that engages the inhibitory peptides in a clamp-like structure. Peptide binding promotes a conformation of a glycine-rich loop that is associated with catalytic activity in other protein kinases. The LRP6 peptide complexes support a model in which phosphorylated receptor acts to inhibit GSK-3 directly and thereby promote β-catenin stabilization. The interactions between GSK-3 and the different inhibitory peptides help to explain substrate sequence preferences. The remodeling of the enzyme around peptide substrates suggests novel avenues for the development of GSK-3 inhibitors specific to particular pathways.

## Results

### A phosphorylated LRP6 repeat motif inhibits GSK-3 more strongly than the phosphorylated GSK-3 N-terminus

Inhibition constants of approximately 700 μM and 13 μM have been reported for the pS9 auto-inhibitory peptide and LRP6 a-motif peptide (NPPPpSPApTERSH), respectively, but these were measured with different substrates and with different assays (*Dajani et al., 2001*; *Piao et al., 2008*). To compare the inhibitory activity of these two peptides directly in the same experimental system, we used a primed peptide substrate from eIF2B (peIF2B) that contains a single phosphorylation site, to avoid complications arising from processive phosphorylation of multiple sites present in many GSK-3 substrates, including glycogen synthase and β-catenin. We also used a longer GSK-3β pS9 peptide than used in *Dajani et al. (2001)*, residues 3–15 (GRPRTTpSFAESCK): Glu12 is conserved, and we wanted to examine its potential contribution to inhibition in the context of the larger peptide.

The doubly-phosphorylated LRP6 a-motif peptide inhibits GSK-3 with a $K_i$ = 1.4 ± 0.2 μM, 40x better than the pS9 peptide (60 ± 11 μM; *Figure 1A,B*; *Table 1*). The 12-fold decrease in $K_i$ measured for the pS9 peptide compared to previously reported values might be due to the use of different substrate peptides, but might also be the result of using the longer pS9 peptide. The 40-fold stronger $K_i$ for the LRP6 peptide compared to the pS9 peptide in the same experimental system supports the conclusion that the LRP5/6 motifs can act as direct GSK-3 inhibitors (*Piao et al., 2008*).

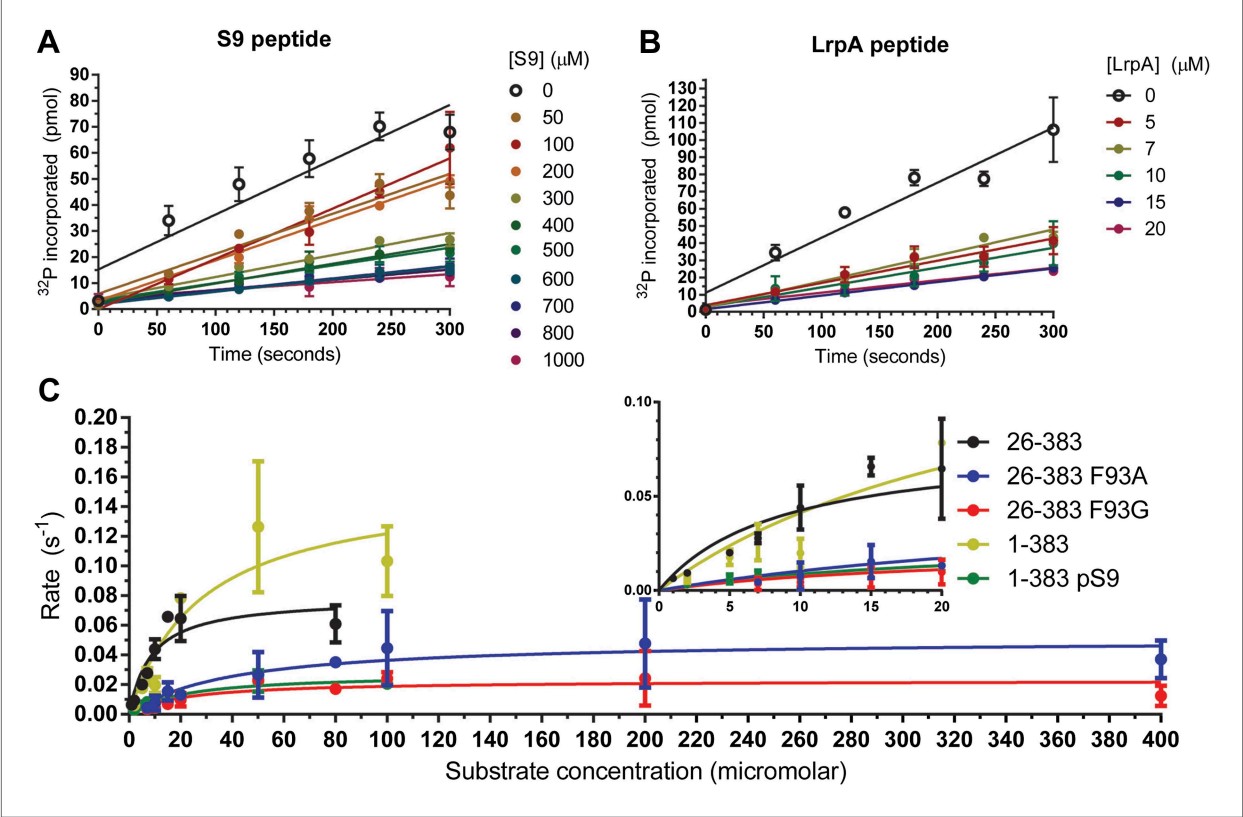

**Figure 1**. Inhibitory activities of pS9 and LRP6 a-motif peptides. (**A** and **B**) [32]P incorporation into the peIF2b substrate over time for GSK-3β 26-383, inhibited by the S9 peptide ($K_i$ = 60 ± 11 μM) (**A**) or the LRP6 a-motif peptide ($K_i$ = 1.4 ± 0.2 μM). The number of replicates is either two or four for each inhibitor concentration, except for the 15 μM LrpA peptide timecourse, where there is only one measurement. (**C**) Determination of steady-state kinetic constants for GSK-3β 26-383, GSK-3β 26-383[F93A], GSK-3β 26-383[F93G], GSK-3β 1-383, and GSK-3β 1-383[pS9]. Error bars represent the standard error of the mean. The number of replicates for each concentration is between 2 and 4 for all the concentration points, except for the 1 μM point of GSK-3 26-383, the 80 μM point of GSK-3β 26-383[F93A], and the 50 and 80 μM points of GSK-3β 26-383[F93G], which have only one measurement.

## The inhibitory peptides bind to GSK-3 as pseudo-substrates

We determined crystal structures of GSK-3β residues 1-383 phosphorylated at the inhibitory pS9 residue by AKT at 2.1 Å resolution, as well as GSK-3β bound to phosphorylated LRP6 c- and e-motif inhibitory peptides at 2.3 Å resolution (*Table 2*). The LRP6 complexes were produced by replacing the first 12 residues of GSK-3β with the 8-residue c- or e-motif from LRP6. The LRP6 peptides in these chimeric proteins were phosphorylated by the chimeric GSK-3 and by CK1 added to the purified protein. Phosphorylation in dilute solution proved to be very inefficient, so phosphorylation was accomplished during crystallization by adding the appropriate kinase to the wild-type or chimeric GSK-3β and dialyzing against high-molecular weight polyethylene glycol ('Materials and methods'). We confirmed that GSK-3β 1-383 that was partially phosphorylated in this manner by AKT was significantly less active than the non-phosphorylated enzyme (*Figure 1C*; *Table 1*).

All of the structures were obtained as complexes with the GSK-3 binding segment of Axin, which was used to promote solubility during protein expression in *Escherichia coli* and to avoid crystal packing interactions in Axin-free GSK-3 structures that likely block access to the substrate-binding pocket (*Dajani et al., 2001*; *ter Haar et al., 2001*). For direct comparison, we re-determined the crystal structure of the peptide inhibitor-free structure of the GSK-3 (residues 1–383)/Axin complex using the same crystallization protocol for the inhibited complexes, at 2.5 Å resolution (*Table 2*). The overall structure of the GSK-3/Axin complexes is similar to those reported previously (*Dajani et al., 2003*; *Tahtouh et al., 2012*) (*Figure 2A*). In each structure, a molecule of ADP is sandwiched between the N- and C-terminal lobes. ATP was added to the protein preparation prior to crystallization, but may have hydrolyzed during crystallization.

**Table 1.** Kinetic parameters for GSK-3β variants

| | $K_M$ (μM) | $k_{cat}$ (s$^{-1}$, × 10$^{-2}$) | $k_{cat}/k_M$ (M$^{-1}$•s$^{-1}$, × 10$^3$) |
|---|---|---|---|
| GSK-3β 26-383 | 8.5 ± 3.8 | 7.9 ± 1.2 | 9.3 ± 4.4 |
| GSK-3β 26-383[F93A] | 39.1 ± 13.8 | 5.1 ± 0.6 | 1.3 ± 0.5 |
| GSK-3β 26-383[F93G] | 21.3 ± 14.2 | 2.3 ± 0.4 | 1.1 ± 0.8 |
| GSK-3β 1-383 | 27.8 ± 15.7 | 15.6 ± 3.6 | 5.6 ± 3.4 |
| GSK-3β 1-383[pS9] | 21.7 ± 9.0 | 2.7 ± 0.4 | 1.2 ± 0.5 |

Both the pS9 auto-inhibitory N-terminal peptide and the phosphorylated LRP6 motifs occupy the primed substrate binding pocket predicted from the presence of phosphate or sulfonate in earlier peptide-free structures (*Dajani et al., 2001*; *Frame et al., 2001*; *ter Haar et al., 2001*; *Figure 2B–E*). Much of the N-terminus is disordered: in the pS9 N-terminal peptide complex, residues [6]RTTpSF are visible, but only the backbone of Arg6 is visible. In the LRP6 inhibitory peptide complexes, residues [1569]PPPpTPR of the c-motif or [1604]PPPpSPC of the e-motif are visible; the second phosphorylation site in these peptides is disordered and we were not able to ascertain whether it is phosphorylated in the crystallized protein.

The pSer/pThr in the primed P+4 position of all three inhibitors binds to the site predicted from peptide-free structures that contain phosphate or sulfonate groups in this region (*Dajani et al., 2001*; *ter Haar et al., 2001*). Arg96, Arg180 and Lys205 form hydrogen bonds with the phosphate group (*Figure 2C–E*). In the inhibitor-free structure (*Figure 2C*), these basic residues form hydrogen bonds with a molecule of glycerol present in the phosphate-binding site, and a previous structure of GSK-3 bound to a non-hydrolyzable ATP analog AMP-PNP shows that water molecules occupy this site (PDB 1PYX; *Bertrand et al., 2003*). These observations suggest that the primed P+4 phosphate-binding site is essentially pre-formed in the enzyme.

As in other kinases, the activation loop region (residues 200–226) forms part of the binding site for substrate peptides. In addition to the basic side chains noted above, the backbone amide of Val214 also forms a hydrogen bond with the phosphate in the priming site. To accommodate the backbone of the inhibitory peptides, the side chain of Val214 adopts a rotamer that avoids steric clash with the phosphate. Residues in the P+1, P+2, and P+3 positions (Arg-Thr-Thr in the pS9 peptide, or Pro-Pro-Pro in the LRP6 c- and e-motif peptides) form an arch over the conserved Ile217, and the P+1 residue packs against Tyr216 (discussed in more detail below). In addition to these interactions, the LRP6 peptide complexes also reveal a water-mediated connection between the backbone carbonyl oxygens of the P+2 proline and Tyr216.

## The C-loop undergoes a large conformational change upon inhibitory peptide binding

Comparison of the inhibited and peptide substrate-free GSK-3 structures shows that the loop immediately preceding the C-helix, designated here as the C-loop, undergoes a large movement toward the C-terminal lobe of the kinase and clamps down on top of the bound pS9 or LRP6 peptides (*Figure 3A*). The α-carbon of residue Phe93, which lies at the tip of the loop, moves 8.5 Å relative to substrate-free structures (*Figure 3A*). In peptide substrate-free GSK-3 crystal structures, many of the side chains in the C-loop are disordered, and the loop has relatively high temperature factors, indicating conformational flexibility. It should be noted that crystals of GSK-3 grown in the absence of Axin have lattice contacts that would not allow the C-loop to adopt the conformation observed here (e.g., *Dajani et al., 2001*; *ter Haar et al., 2001*).

In all three inhibitory peptide complexes, Phe93 is sandwiched in between the P+2 and P+5 residues of the peptide (*Figures 2B and 3B*). Hydrogen bonds form between the backbones of the P+4 and P+5 residues and the backbone of the C-loop from residues 92–94 (*Figure 3B*). Asp90 also participates in stabilizing the alternate C-loop conformation by hydrogen bonding to the backbone amide of Arg92.

The movement of the C-loop toward the peptide is associated with a small 'closing' movement (rmsd 0.5 Å, 0.6° rotation, not including the C-loop or glycine-rich loop) of the N-terminal lobe toward the C-terminal lobe relative to the non-substrate bound structures of GSK-3 (*Figure 3A*). Directly

**Table 2.** X-ray crystallography data collection and refinement statistics

| | Inhibitor-free | pS9 | pS9-AlF$_3$ | LRP6 c-motif | LRP6 e-motif |
|---|---|---|---|---|---|
| **PDB code** | **4NM0** | **4NM3** | **4NU1** | **4NM5** | **4NM7** |
| Data collection* | | | | | |
| Space group | $P6_122$ | $P6_122$ | $P6_122$ | $P6_122$ | $P6_122$ |
| Unit cell lengths $a$, $c$ (Å) | 81.3, 280.8 | 81.0, 281.1 | 81.0, 280.5 | 81.7, 280.9 | 82.0, 280.3 |
| Beamline | SSRL 11-1 | SSRL 11-1 | APS 23-ID-B | SSRL 12-2 | SSRL 12-2 |
| Wavelength (Å) | 1.033 | 1.033 | 1.033 | 1.00 | 1.00 |
| Resolution range (Å) | 39.1–2.50 | 39.0–2.10 | 46.8–2.50 | 39.2–2.30 | 39.4–2.30 |
| (last shell) | (2.60–2.50) | (2.16–2.10) | (2.60–2.50) | (2.38–2.30) | (2.38–2.30) |
| Unique reflections | 20,069 | 33,093 | 19,922 | 25,685 | 25,932 |
| $CC_{1/2}$† | 0.999 (0.726) | 1.00 (0.673) | 0.999 (0.814) | 0.999 (0.459) | 1.00 (0.485) |
| $R_{merge}$‡ | 0.106 (1.74) | 0.100 (4.88) | 0.303 (6.772) | 0.141 (3.63) | 0.136 (4.31) |
| $<I>/<\sigma I>$ | 18.4 (1.2) | 24.5 (0.8) | 13.0 (0.7) | 18.9 (1.0) | 17.8 (0.9) |
| Completeness (%) | 100 (99.9) | 99.9 (99.7) | 100 (100) | 99.7 (99.6) | 100 (100) |
| Multiplicity | 9.6 (9.9) | 18.6 (17.9) | 21.0 (21.6) | 18.9 (18.1) | 18.7 (18.2) |
| Refinement | | | | | |
| No. reflections work/test set | 19,958/979 | 32,924/1658 | 19,823/974 | 25,608/1283 | 25,838/1292 |
| $R_{work}$/$R_{free}$§ | 0.191/0.240 | 0.194/0.242 | 0.191/0.245 | 0.183/0.232 | 0.183/0.234 |
| Number of atoms | | | | | |
| GSK-3 | 2780 | 2885 | 2862 | 2850 | 2858 |
| Axin | 165 | 153 | 149 | 155 | 155 |
| LrpC/E peptide | – | – | – | 44 | 44 |
| ADP | 27 | 27 | 27 | 27 | 27 |
| Mg$^{2+}$ | 2 | 2 | 2 | 2 | 2 |
| Cl$^-$ | 1 | 1 | – | 1 | 1 |
| Glycerol | 30 | 24 | 12 | 24 | 18 |
| DTT | 8 | 8 | – | – | 8 |
| AlF$_3$ | – | – | 4 | – | – |
| NO$_3^-$ | – | – | 4 | – | – |
| Water | 164 | 197 | 98 | 121 | 106 |
| *B*-factors (Å$^2$) | | | | | |
| GSK-3 | 52.6 | 55.0 | 75.1 | 69.7 | 69.7 |
| Axin | 59.5 | 54.0 | 78.1 | 75.2 | 77.8 |
| LrpC/E peptide | – | – | – | 101 | 100 |
| ADP | 65.3 | 50.3 | 57.5 | 72.2 | 56.5 |
| Mg$^{2+}$ | 88.5 | 53.8 | 57.6 | 86.4 | 69.6 |
| Cl$^-$ | 77.1 | 71.1 | – | 90.9 | 96.8 |
| Glycerol | 65.2 | 79.4 | 93.1 | 94.6 | 97.2 |
| DTT | 99.3 | 102 | – | – | 131 |
| AlF$_3$ | – | – | 83.6 | – | – |
| NO$_3^-$ | – | – | 100.8 | – | – |
| Water | 46.4 | 52.3 | 61.1 | 64.3 | 64.6 |
| Rmsd | | | | | |
| Bond lengths (Å) | 0.003 | 0.005 | 0.003 | 0.002 | 0.004 |
| Bond angles (°) | 0.63 | 0.91 | 0.63 | 0.63 | 0.76 |

*Table 2. Continued on next page*

*Table 2. Continued*

| | Inhibitor-free | pS9 | pS9-AlF$_3$ | LRP6 c-motif | LRP6 e-motif |
|---|---|---|---|---|---|
| **PDB code** | **4NM0** | **4NM3** | **4NU1** | **4NM5** | **4NM7** |
| Ramachandran plot (%)¶ | | | | | |
| Favored regions | 96.5 | 96.9 | 96.1 | 95.6 | 95.8 |
| Additional allowed regions | 3.5 | 3.1 | 3.9 | 3.9 | 4.2 |
| Outliers | 0 | 0 | 0.5 | 0.5 | 0 |

*Values in parentheses are for highest-resolution shell. Rmsd, root mean square deviation.

†As defined in Aimless (***Evans and Murshudov, 2013***).

‡$R_{merge} = \Sigma_h\Sigma_i|I_i(h)-< I(h) >|/\Sigma_h\Sigma_i(h)$, where $I_i(h)$ is the $I^{th}$ measurement of reflection h, and $< I(h) >$ is the weighted mean of all measurements of h.

§$R = \Sigma_h|F_{obs}(h)-F_{calc}(h)|/\Sigma_h|F_{obs}(h)|$. $R_{work}$ and $R_{free}$ were calculated using the working and test reflection sets, respectively.

¶As defined in MolProbity (***Chen et al., 2010***).

adjacent to the C-loop, Leu88 is pulled toward the inhibitory peptide, and Tyr127 moves downward and sits on top of Leu88 (***Figure 3B***). These movements are transmitted along the corresponding β strands to which these residues attach, which also shift slightly (***Figure 3B***). This shift of the β-sheet includes the glycine-rich loop that sits above the nucleotide.

## The inhibitor peptides promote a catalytically competent conformation of the glycine-rich loop

In crystal structures of GSK-3 lacking an inhibitory peptide, the glycine-rich loop of the N-terminal lobe, comprising residues 62–70, adopts a position similar to that observed in the crystal structure of PKA bound to AMP (***Narayana et al., 1997***; ***Figure 4A,B***) In contrast, in both the pS9 peptide and LRP6 e-motif complexes we observe an additional, partially occupied conformation of the glycine-rich loop in which the loop has moved toward the ADP molecule and away from the rest of the N-terminal lobe, with the backbone amide of Ser66 moving by 4 Å compared to the peptide-free structure (***Figure 4A,C***). This lower conformation could not be modeled in the LRP6 c-motif complex, although the refined Fo-Fc electron density suggests that it may be present at very low occupancy. In the pS9 complex, both Ser66 and Phe67 now adopt positions nearly identical to those of Ser53 and Phe54 of PKA bound to the transition state analog ADP+AlF$_3$ and a peptide (***Madhusudan et al., 2002***; ***Figure 4B***). Phe67 packs against the threonine at the P+2 residue of the pS9 peptide, and although Ser66 and Phe67 are poorly ordered in the LRP6 c- and e-motif complexes, modeling them in the same position as that observed in the pS9 complex shows that Phe67 can pack against the P+2 proline of the LRP6 peptides. Mutation of GSK-3β Phe67 to alanine severely impairs catalytic activity (***Ilouz et al., 2006*** and MDE, data not shown), suggesting that the packing of this residue against the substrate peptide is important for stabilizing a catalytically active conformation.

To confirm that the lower glycine-rich loop conformation corresponds to a catalytically competent conformation, we soaked crystals of the pS9 complex in AlF$_3$. The trigonal planar AlF$_3$ molecule is coordinated by an ADP β-phosphate oxygen, as predicted for AlF$_3$ mimicking the transition state of an in-line attack on the Ser/Thr residue of the substrate (***Madhusudan et al., 2002***), and the lower conformation of the glycine-rich loop is well defined (***Figure 4C***). Despite the slightly lower resolution of this structure (***Table 2***), the glycine-rich loop adopts the same conformation as that in the pS9 and LRP6 e-motif complexes, indicating that the presence of ADP and peptide inhibitor stabilize this conformation.

The structural and mutational data strongly suggest that the inhibitory peptides bind as true pseudo-substrates and stabilize a catalytically competent conformation. The partial occupancy of the lower glycine-rich loop conformation, along with the disorder of residues 65–67 in the LRP6 complexes, is consistent with NMR studies of PKA, which show that the glycine-rich loop is relatively ordered in the upper conformation in the absence of peptide substrate, whereas it becomes more dynamic when the peptide is bound (***Masterson et al., 2010***, ***2011***).

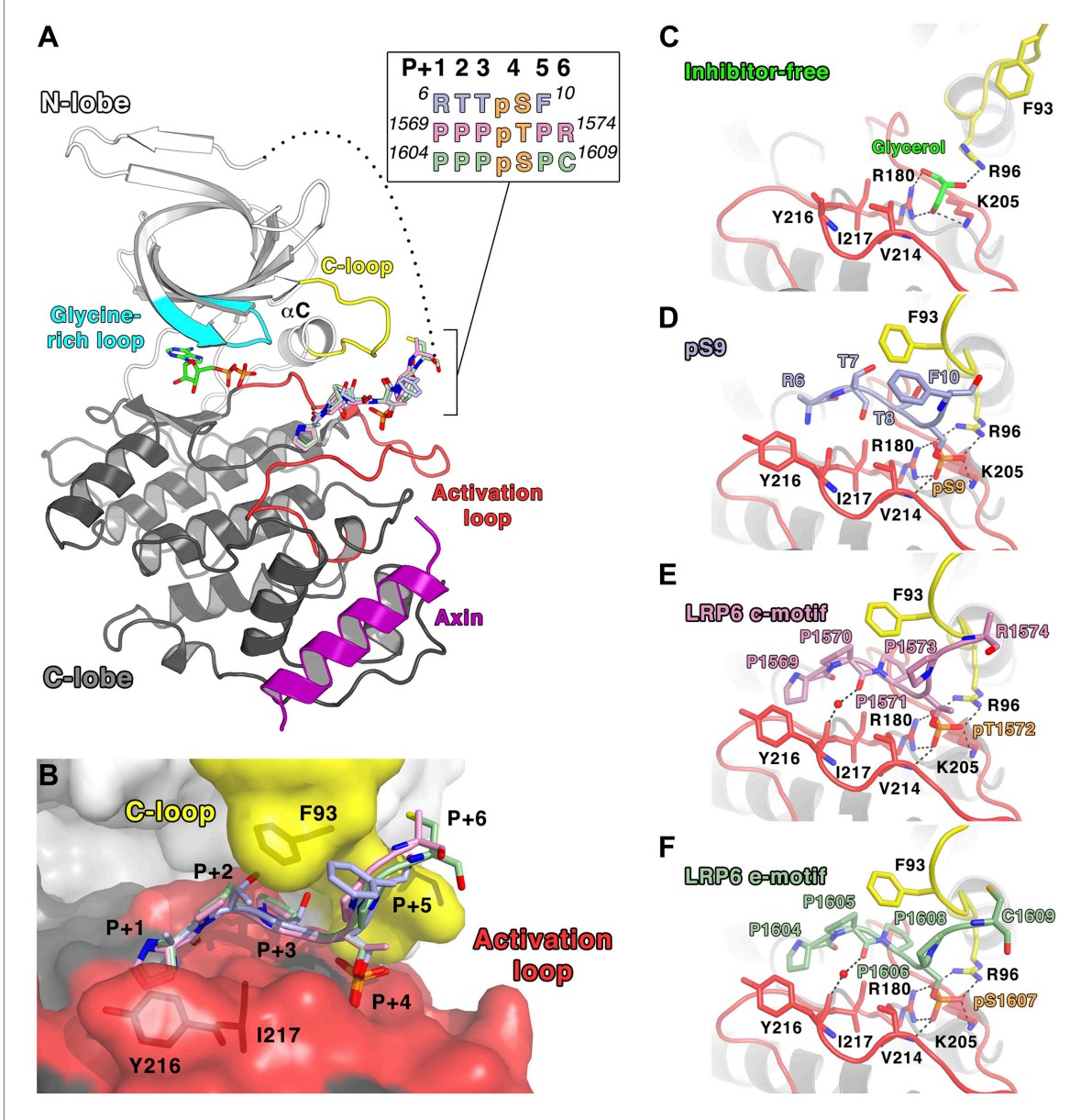

**Figure 2**. Inhibitory peptide binding to GSK-3. (**A**) Overall structure of GSK-3 bound to inhibitory peptides. The superimposed LRP6 c-motif (pink sticks), e-motif (light green sticks) and pS9 auto-inhibitory N-terminal peptide (light blue sticks) bind to the same substrate-binding pocket between the C-loop (yellow) and activation loop (red). A molecule of ADP binds to the deep cleft located between the N-terminal (white) and C-terminal (grey) lobes, and the Axin helix (purple) binds at the C-lobe. The glycine-rich loop (cyan) and αC-helix are also indicated. The inset shows the protein sequences of the peptide residues that are visible in the structures. The P+4 phosphorylated residues are indicated in orange. The loop between the N-terminal peptides and the first β strand of the N-terminal lobe is partially disordered (dotted line). Oxygen atoms are shown in red, nitrogen in blue, phosphorus in orange, and sulfur in yellow. (**B**) Surface representation of the substrate-binding pocket between the C-loop (yellow) and activation loop (red) of GSK-3. The inhibitory peptides, pS9 auto-inhibitory N-terminal peptide (light blue sticks), LRP6 c-motif (pink sticks) and e-motif (light green sticks) are superimposed, and the residues of the peptides are labeled according to the primed substrate numbering, with the phospho-serine or threonine at the P+4 position. Side chains of GSK-3 residues F93, Y216 and I217, which interact with the peptides, are also depicted as sticks. (**C**) Peptide inhibitor-free structure near the C-loop and activation loop. A molecule of glycerol is bound to three basic residues that interact with the phosphate at the substrate P+4 site. Hydrogen bonds are shown as dashed lines. (**D**–**F**) Interactions between GSK-3 and inhibitory peptides. The structural water molecules that interact between the carbonyl groups of Y216 and the P+1 proline residues of LRP6 c-motif and e-motif peptides are depicted as red spheres.

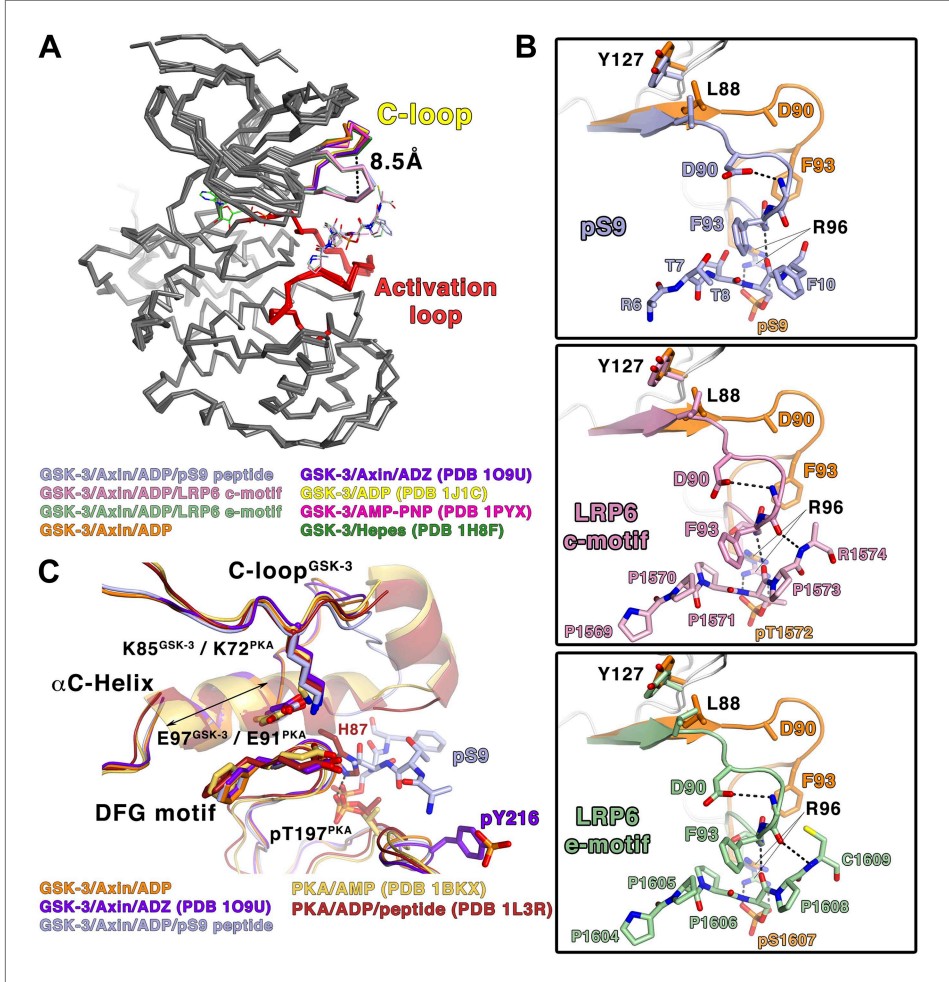

**Figure 3**. Conformational changes in GSK-3 upon inhibitor peptide binding. (**A**) Superposition of α-carbon traces of GSK-3 structures. The maximum deviation between the structures is at the C-loop region, which is color coded according to the key at the bottom of the panel. The dashed line indicates the maximum distance (8.5 Å) between the α-carbon of residue Phe93 seen in peptide substrate-bound and peptide-free structures. The activation loops (red) align very well, indicating that there is no conformational change at the activation loop upon peptide substrate binding. (**B**) Changes of the C-loop upon peptide substrate binding. Superposition of the peptide substrate-free GSK-3 (inhibitor-free; *Table 2*) (orange) and the GSK-3/pS9 peptide (top panel), LRP6 c-motif (middle panel), or LRP6 e-motif (bottom panel) complexes. Hydrogen bonds between the backbones of the P+4 and P+5 residues of the peptides and the backbone of the C-loop from R92 to K94 are shown as dashed lines; only the main-chain atoms of R92 and K94 are depicted. The change in the C-loop upon peptide binding alters the positions of the preceding β strands (broad arrows). R96 adopts a conformation to accommodate the P+4 substrate phosphate. (**C**) Superposition of GSK-3 and active PKA structures. The catalytic residues, including K85, E97 and the DFG motif, of the peptide-free GSK-3 (orange), peptide-free GSK-3 with phosphorylated Y216 (purple) or peptide-bound GSK-3 (light blue) adopt similar conformations as those of the active PKA structures (gold/brown). The double-headed arrow indicates the shorter length of the αC-helix of GSK-3 relative to that of PKA. The hydrogen bond between phosphorylated T197 (pT197) and H87 is shown as dashed line; only H87 of the PKA/peptide complex is depicted for clarity. pT197 of PKA places a phosphate group in the space equivalent to that occupied by the substrate phosphate in GSK-3.

In the structures of GSK-3 bound to both ADP and pseudo-substrate peptides, the formation of the lower glycine-rich loop conformation occurs in the absence of a γ-phosphate on ATP or a transition-state mimic, but appears to be further stabilized by AlF₃ (*Figure 4B,C*). The interaction between Phe67 and the P+2 residue of a substrate, and the movement of the β strand adjacent to the glycine-rich loop that occurs when the C-loop engages the substrate peptide, may favor the lower conformation of the glycine-rich loop seen here. For example, Phe67 is found in several positions in peptide inhibitor-free

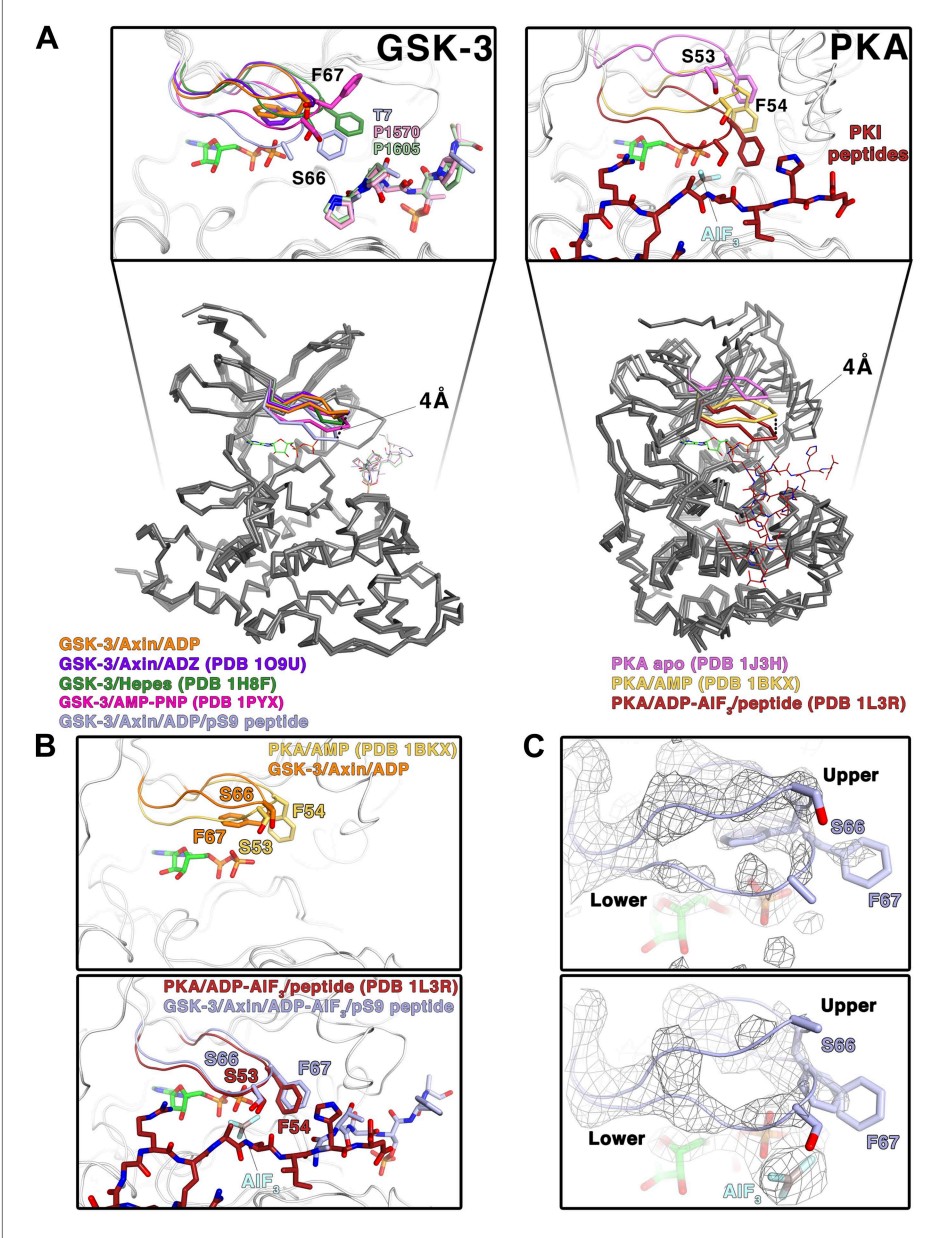

**Figure 4**. Conformational change of the glycine-rich loop upon inhibitor peptide binding. (**A**) *Left panel*: superposition of α-carbon traces of five selected GSK-3 structures. The dashed line indicates the maximum deviation at the glycine-rich loops among the currently solved GSK-3 structures (4 Å at the α-carbon of residue S66 between the peptide-free (orange/purple) and the peptide-bound (light blue) GSK-3). Some GSK-3 structures show an intermediate position of the glycine-rich loop (green/pink) due to interaction with another GSK-3 molecule in the crystal lattice (not shown in the figure for clarity, please refer to PDB 1H8F/1PYX). The inset shows the close-up of the glycine-rich loops of the five selected GSK-3 structures. Upon peptide binding, F67 packs against the T7 or P1570/P1605 at the P+2 residue of the pS9 or LRP6 peptides. *Right panel*: superposition of α-carbon traces of three selected PKA structures. The glycine-rich loop of the nucleotide-free PKA (light purple) is at the 'highest' position, due to the open conformation of the inactive kinase structure. In the nucleotide-bound state (gold), the glycine-rich loop moves toward the C-terminal lobe, and it moves further down by 4 Å upon binding to a peptide in its substrate-binding site (brown). The inset shows the close-up of the glycine-rich loops of the three selected PKA structures. The equivalent residues of GSK-3 S66 and F67, PKA S53 and F54, are depicted as sticks. The ADP and AlF$_3$ molecules are also shown. (**B**) Close-up comparisons of the glycine-rich loops in the nucleotide-bound state (upper panel) and nucleotide/peptide-bound transition state (lower panel) of GSK-3 and PKA. (**C**) Comparisons of

*Figure 4. Continued on next page*

*Figure 4. Continued*
the electron density (gray mesh, 2Fo-Fc map contoured at 0.8 σ) of the glycine-rich loops in the ADP-bound state (upper panel) and ADP–AlF$_3$ transition state (lower panel). The upper and lower conformations of the glycine-rich loops are indicated. GSK-3 S66 and F67 are depicted as sticks, and the ADP and AlF$_3$ molecules are also shown.

GSK-3 structures, and some of them (e.g., PDB 1PYX) would clash with the C-loop in the peptide-bound structures (*Figure 4A*). We conclude that binding of the pseudo-substrate itself favors the lower glycine-rich loop conformation in GSK-3, which strongly suggests that true substrates also favor this loop conformation and thereby promote catalysis (summarized in *Video 1*). Unlike PKA, however, substrate binding requires a significant conformational rearrangement of the C-loop in the N-terminal lobe.

The phosphorylated residue of the inhibitory peptides binds to the predicted P+4 binding pocket of primed GSK-3 substrates (*Dajani et al., 2001*; *Frame et al., 2001*; *ter Haar et al., 2001*), and the position of the phosphate corresponds closely to that of phosphorylated Ser197 in the activation loop of PKA (*Figure 3C*). However, we do not observe the P+0 residue in any of our structures, which leaves open the possibility that the observed binding mode is distinct from that of true substrates. We compared the orientation of the observed residues by superposition of several known Ser/Thr kinase:peptide substrate complex structures in which the P+0 residue is observed. The P+1 residue in these complexes, as well as that of the PKA inhibitor PKI, aligns very well with those of the GSK-3 inhibitory peptides, and is oriented similarly such that the P+0 residue would be correctly positioned to accept a phosphate from ATP (*Figure 5*). Combined with the effect of the conformation of the glycine-rich loop, it appears that the observed binding mode of the inhibitory peptides corresponds closely to the binding mode of true GSK-3 substrates.

## Phe93 may guide substrate sequence specificity for the P+2 and P+5 positions

Approximately 50% of the total buried surface area of each inhibitory peptide is involved in interactions with the C-loop, suggesting that the C-loop is a key element of target recognition specificity. In addition to the main-chain interactions observed between GSK-3 and the inhibitory peptides, GSK-3β Phe93 packs against residues at the P+2 and P+5 positions of the inhibitor peptides. Given that the inhibitor peptides appear to bind as pseudo-substrates and stabilize an active conformation of GSK-3, we tested the importance of Phe93 for GSK-3 activity by determining steady-state kinetic parameters for wild-type GSK-3β and the F93A and F93G mutants using peIF2b as the substrate (*Table 1*; *Figure 1C*). The mutants show 7–9-fold reduction in catalytic efficiency ($k_{cat}/K_M$), with significant increases in $K_M$ that suggest a loss of affinity for the peptide substrate. These data indicate that the interaction with Phe93 contributes strongly to substrate binding, and the reductions in $k_{cat}$ are consistent with the notion that peptide binding is required for formation of a catalytically competent enzyme.

Unlike most loops in the N-terminal lobe of GSK-3, the C-loop has been highly conserved throughout evolution (*Figure 6*). The position equivalent to Phe93 of mouse GSK-3β is strongly conserved; the only other residue found at this position is tyrosine, which would be able to form the same interactions with the P+2 and P+5 residues of the substrate peptide. A bulkier residue such as tryptophan at this position would clash with the substrate peptide backbone, and the decreased activity of the F93A mutant suggests that smaller residues would not extend far enough to interact with the substrate. In the absence of pseudo-substrate inhibitors, Phe93 does not interact with other parts of the kinase and is present on a solvent exposed, flexible loop.

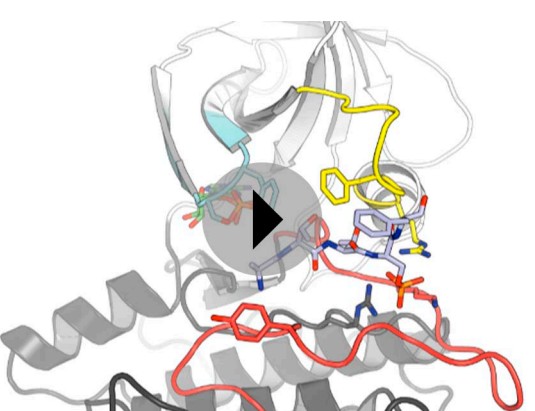

**Video 1**. Conformational changes in GSK-3 associated with peptide substrate binding. The video shows the ADP and Axin-bound enzyme changing between the inhibitor-free and pS9-bound states. Key structural elements are colored as in *Figure 2A*.

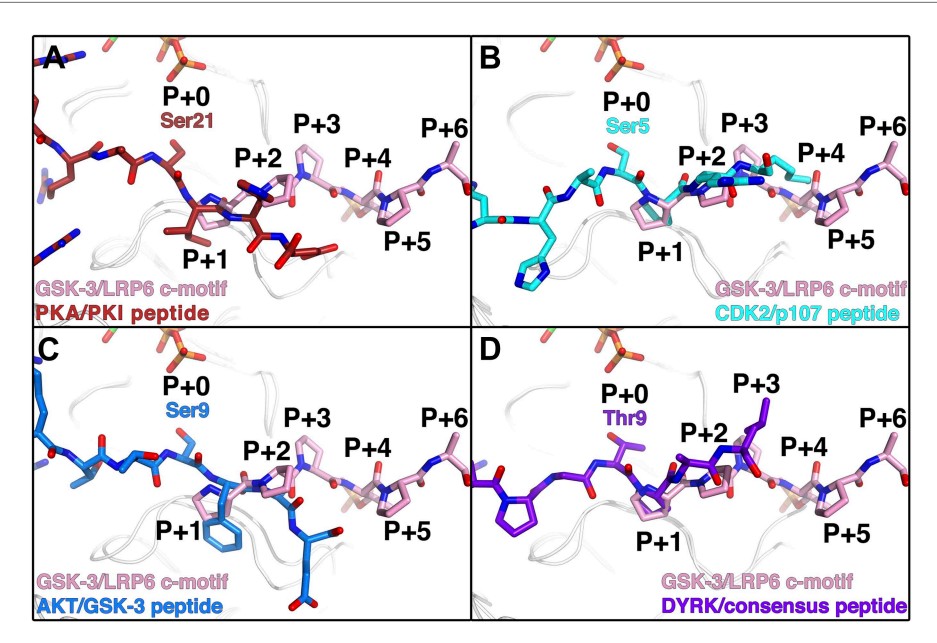

**Figure 5**. Comparison of GSK-3 inhibitory peptide orientation with other kinase: peptide complexes. Superposition of the GSK-3/LRP6 c-motif structure with the (**A**) PKA:PKI peptide (PDB 1L3R); (**B**) CDK2:p107 peptide (PDB 1QMZ); (**C**) AKT:GSK-3 peptide (PDB 3CQU); and (**D**) DYRK:consensus peptide (PDB 2WO6) substrate complexes. The β-phosphate of the ADP molecule in the catalytic site of GSK-3 is shown at the top of each panel, with phosphorus in orange and oxygen in red.

Thus, the conservation of Phe or Tyr residue at this position must be attributed to its role in substrate binding and concomitant formation of an active kinase conformation.

The interactions of the P+2 site with Phe67 and Phe93, and the packing of the P+5 residue against Phe93, are consistent with the sequence preferences of these positions seen in experimentally confirmed GSK-3 substrates (*Sutherland, 2011*; *Table 3*; *Figure 7A*). The P+2 position has 43% hydrophobic and

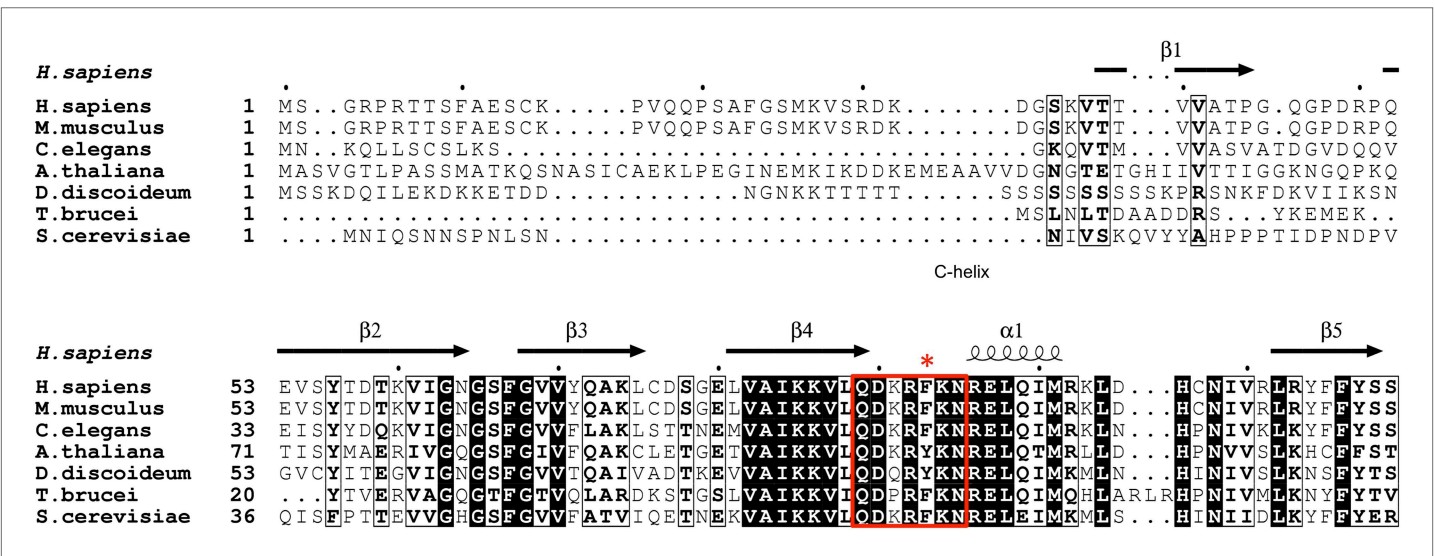

**Figure 6**. GSK-3 sequence alignments near the C-loop. White letters on black background indicates identical residues, and bold letters denote similar residues. Secondary structure elements are shown above the alignment. The C-loop is marked in the red box, with Phe93 indicated by the asterisk.

**Table 3.** Confirmed GSK-3 biological targets (adapted from *Sutherland, 2011*)

| Protein | P+0 residue | P+0 to P+5 sequence | UniprotKB # |
|---|---|---|---|
| APC | 1501 | SCSSSL | (P25054) |
| ATP-citrate lyase | 447 | TPAPSR | (P53396) |
| | 451 | SRTASF | |
| Axin | 317 | SANDSE | (O15169) |
| | 321 | SEQQSL | |
| BCL-3 | 398 | SPSSSP | (P20749) |
| | 402 | SPSQSP | |
| β-catenin | 33 | SGIHSG | (P35222) |
| | 37 | SGATTT | |
| | 41 | TTAPSL | |
| C/EBPα | 226 | TPPPTP | (P49715) |
| | 230 | TPVPSP | |
| C/EBPβ | 223 | SLSTSS | (P17676) |
| | 227 | SSSSSP | |
| | 231 | SPPGTP | |
| Ci155 (Gli3) | 861 | SRRSSG | (P10071) |
| | 873 | SRRSSE | |
| | 903 | SRRSSE | |
| CLASP2 | 533 | SRESSR | (O71522) |
| | 537 | SRDTSP | |
| CRMP2 | 514 | TPASSA | (Q16555) |
| | 518 | SAKTSP | |
| CRMP4 | 514 | TPAGSA | (Q14195) |
| | 518 | SARGSP | |
| CREB | 129 | SRRPSY | (P16220) |
| CRY2 | 554 | SGPASP | (Q49AN0) |
| Cytidine triphosphate synthetase | 571 | SGSSSP | (P17812) |
| Dynamin I | 776 | TSSPTP | (Q05193) |
| eIF2B | 540 | SRGGSP | (Q13144) |
| FAK | 722 | SPRSSE | (Q05397) |
| Glycogen Synthase | 641 | SVPPSP | (P13807) |
| | 645 | SPSLSR | |
| | 649 | SRHSSP | |
| | 653 | SPHQSE | |
| Heat shock factor 1 | 303 | SPPQSP | (Q00613) |
| HIF1α | 551 | STQDTD | (Q16665) |
| hnRNP D | 83 | SPRHSE | (Q14103) |
| IRS1 | 337 | SRPASV | (P35568) |
| c-jun | 239 | TPPLSP | (P05412) |
| MAP1B | 1396 | SPLRSP | (P46821) |
| Mcl1 | 159 | SLPSTP | (Q07820) |
| Mdm2 | 242 | SDQFSV | (Q00987) |
| | 256 | SEDYSL | |
| MLK3 | 789 | SPLPSP | (Q16584) |

*Table 3. Continued on next page*

*Table 3. Continued*

| Protein | P+0 residue | P+0 to P+5 sequence | UniprotKB # |
|---|---|---|---|
| c-myc | 58 | TPPLSP | (P01106) |
| Myocardin | 451 | STSSSP | (Q8IZQ8) |
| | 455 | SPPISP | |
| | 459 | SPASSD | |
| | 626 | STFLSP | |
| | 630 | SPQCSP | |
| | 634 | SPQHSP | |
| NDRG1 | 342 | SRSHTS | (Q92597) |
| p130Rb | 948 | SHQNSP | (Q08999) |
| | 962 | SRDSSP | |
| | 982 | SAPPTP | |
| p53 | 33 | SPLPSQ | (P04637) |
| PITK | 1007 | SKTVSF | (O15084) |
| Polycystin-2 | 76 | SPPLSS | (Q13563) |
| Presenilin-1 | 397 | SATASG | (P49768) |
| | 353 | STPESR | |
| PP1 G-subunit | 38 | SPQPSR | |
| | 42 | SRRGSE | |
| PTEN | 362 | STSVTP | (P60484) |
| | 366 | TPVDSD | |
| Snail | 96 | SGKGSQ | (O95863) |
| | 100 | SQPPSP | |
| SREBP1a | 426 | TPPPSD | (P36956) |
| | 430 | SDAGSP | |
| Tau | 525 | SRSRTP | (P10693) |
| | 548 | TPPKSP | |
| | 713 | SPVVSG | |
| | 717 | SGDTSP | |
| TSC2 | 1379 | SQPLSK | (P49815) |
| | 1383 | SKSSSS | |
| VDAC | 51 | TTKVTG | (P21796) |
| von Hippel-Lindau | 68 | SREPSQ | (P40337) |

Numbers listed are residue numbers of the P+0 residue of the human proteins, with sequences representing the P+0 through P+5 residues for each target sequence. UniprotKB accession numbers are listed to the right.

79% uncharged residues, consistent with a relatively hydrophobic environment created by the two phenylalanines. The P+5 position has 63% hydrophobic and 81% uncharged residues, with a strong preference for proline, which makes extensive interactions with Phe93 in the LRP6 peptide complexes. Interestingly, only 5 of the 75 targets in *Table 3* contain charged residues at both the P+2 and P+5 positions. Also, tryptophan is completely excluded at these positions, and phenylalanine and tyrosine only appear four times total out of the 75 sequences listed. We speculate that large aromatic residues might interact too strongly with Phe93 and thereby reduce turnover of the enzyme. Alternatively, at least in the P+2 position, large aromatic residues may sterically clash with Phe93 or the nearby Tyr216 or Phe67 residues.

## Role of Tyr216 in substrate binding and sequence specificity

The GSK-3 prepared for crystallography was purified from *E. coli*, and we do not see evidence of phosphorylation of Tyr216 by mass spectrometry of the purified protein (data not shown) or in the

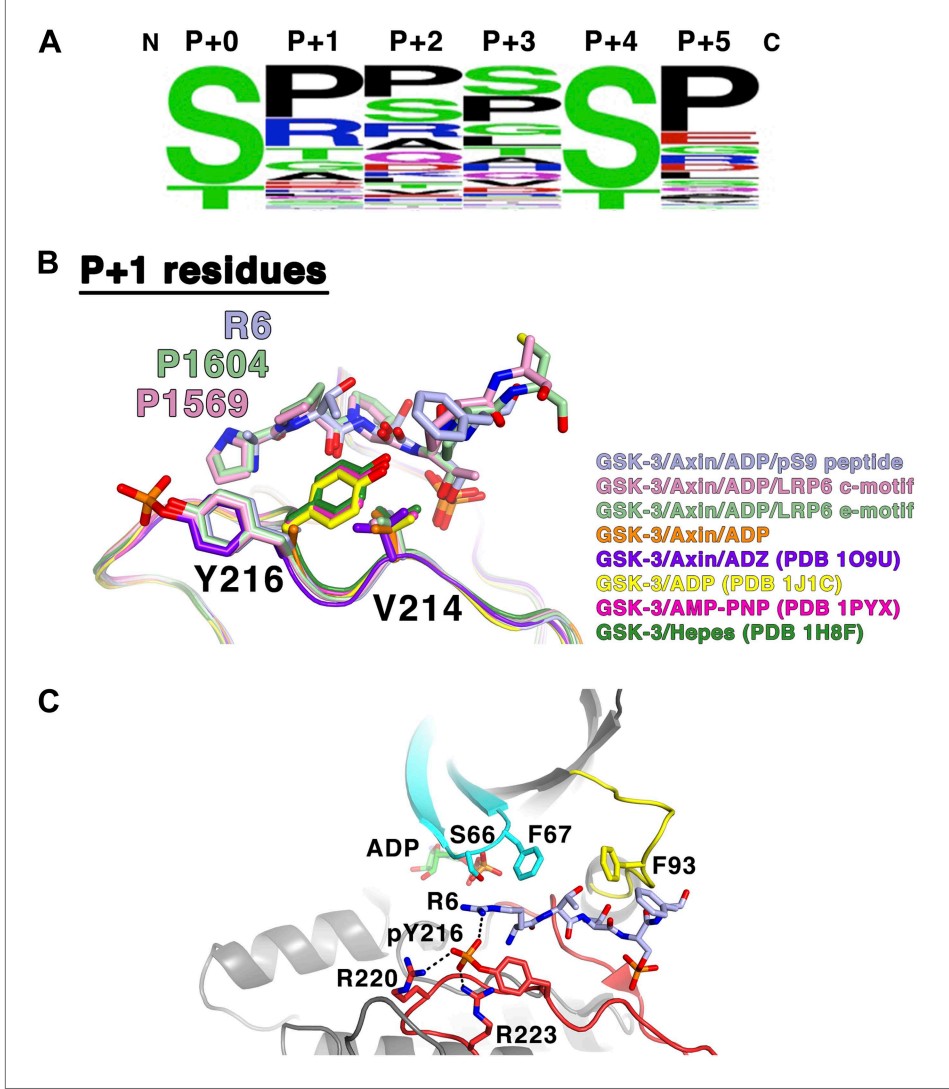

**Figure 7**. GSK-3 substrate specificity. (**A**) Amino acid frequency logo diagram for each of the positions P+0 through P+5 for the 75 GSK-3 substrates listed in *Table 3*. The diagram was created using WebLogo (*Crooks et al., 2004*). (**B**) Conformational change of Y216 upon peptide binding. In the absence of peptide and Y216 phosphorylation, the ring of Y216 packs against V214. In the presence of peptide, Y216 points outward, away from the peptide, and adopts an identical conformation to that seen when it is phosphorylated (purple, GSK-3/Axin/ADZ structure; PDB 1O9U [*Dajani et al., 2003*]). The ring of Y216 packs against the P+1 residue in each of the peptide complexes. (**C**) Model of basic P+1 residue interaction with phosphorylated Y216. The phosphate group of phosphorylated Y216 could interact with the positively charged moiety of arginine at P+1 of the peptide substrate. As seen in PDB 1O9U, Arg220 and Arg223 would also chelate the phosphate group. Also shown are peptide-binding residues on the glycine-rich loop (cyan), C-loop (yellow) and activation loop (red).

electron density maps. Although partial phosphorylation of this residue has been observed previously in material produced in *E. coli* (*Wang et al., 1994*), it may be that the presence of the bulky GST fused to the N-terminus of the Axin peptide, which is located near the GSK-3 active site, precludes GSK-3 from carrying out phosphorylation during expression by sterically blocking the GSK-3 active site.

Crystal structures of GSK-3 containing non-phosphorylated Tyr216 show that the tyrosine side chain points toward the N-terminal lobe, with the tyrosine ring packing against Val214 (*Figure 7B*; *Dajani et al., 2001*; *ter Haar et al., 2001*). The presence of the pS9 or LRP6 peptides, however, precludes this positioning for Tyr216. Instead, it points outward, away from the peptide, and adopts an identical conformation to that seen when it is phosphorylated, where it interacts with several basic residues

(*Bax et al., 2001*; *Dajani et al., 2003*; *Figure 7B*). In the inhibitory peptide complexes, the ring of Tyr216 packs against the P+1 residue. These observations are consistent with the modest increase in GSK-3 activity when Tyr216 is phosphorylated: in order to accommodate binding of the substrate peptide, Tyr216 must move away from its unphosphorylated position and break its interaction with Val214. Having phosphorylated Tyr216 pre-positioned for substrate binding would not require the extra energy needed for this rearrangement.

Alignment of known GSK-3 primed substrates demonstrates some preference for small, uncharged residues, especially proline, at the P+1 position (*Table 3*, *Figure 7A*), consistent with the non-polar interactions observed with the ring of Tyr216. However, arginine is found in this position in 19% of the substrates, including the inhibitory pS9 peptide, where its β carbon packs against Tyr216. When GSK-3 is phosphorylated at Tyr216, the charged phosphate group on pTyr216 would be located near the distal end of the P+1 residue. Modeling shows that the phosphate group could interact with the positively charged moiety of arginine at P+1 (*Figure 7C*). These observations suggest that GSK-3 substrate specificity may be controlled in part through Tyr216 phosphorylation status, with substrates containing smaller or hydrophobic residues at P+1 preferred when Tyr216 is not phosphorylated and basic residues preferred when phosphorylated. If so, potential tyrosine kinases mediating Tyr216 phosphorylation or factors that promote autophosphorylation of Tyr216 may favor phosphorylation of particular substrates and thereby provide an additional level of regulation of GSK-3 function (*Lochhead et al., 2006*; *Kaidanovich-Beilin and Woodgett, 2011*).

## Discussion

The data presented here reveal the mode of binding of GSK-3 pseudo-substrate inhibitors, which imply that the catalytically active conformation can form only in the presence of a peptide substrate. Although the C-loop is often poorly ordered in non-peptide-bound structures, it interacts directly with the inhibitory peptides, and the loss of activity resulting from mutation of Phe93 demonstrates the critical nature of this interaction. The C-loop interactions, coupled with changes in the position of the glycine-rich loop, create an arrangement of catalytic residues comparable to those observed in PKA bound to the transition state mimic ADP-AlF$_3$ and a peptide substrate (*Madhusudan et al., 2002*). These observations, as well as comparison to structures of other kinases bound to substrates, indicate that the inhibitory peptides act as true pseudo-substrates that mimic binding to substrate, rather than stabilizing a distinct catalytically inactive conformation of GSK-3.

Phosphorylation of the conserved LRP5/6 cytoplasmic repeat motifs is essential for Wnt signal transduction and is in particular associated with inhibition of GSK-3 activity and subsequent stabilization of the transcriptional co-activator β-catenin (*MacDonald and He, 2012*). Several studies have shown that phosphorylated LRP5/6 can directly inhibit GSK-3 (*Piao et al., 2008*; *Wu et al., 2009*). This suggests a model in which recruitment of the destruction complex to the Wnt-bound receptors occurs when activated Dishevelled binds to Axin through their respective DIX domains. GSK-3 bound to Axin would phosphorylate LRP5/6 and generate its own inhibitor, thereby selectively inhibiting the pool of GSK-3 that phosphorylates β-catenin without affecting other GSK-3 activities (*Piao et al., 2008*). The kinetic and structural data presented here demonstrate that the phosphorylated LRP6 repeat motifs act as direct inhibitors of GSK-3 catalytic activity. The 40-fold stronger inhibitory activity relative to that of phosphorylated GSK-3 N-terminus is consistent with the LRP5/6 motifs working in *trans*, as opposed to the presumed kinetic advantage of having the GSK-3 N-terminal inhibitory substrate part of the same polypeptide chain, although formation of membrane associated 'signalosome' complexes (*Bilic et al., 2007*) likely enhances the probability of encounter between Axin-bound GSK-3 and LRP5/6.

The prolines of the LRP5/6 cytoplasmic tail repeat motifs have been shown to be critical for Wnt signaling in cells (*MacDonald et al., 2008*). The absolute conservation of prolines at the P+2 and P+5 positions of the five individual PPSP motifs of LRP6, as opposed to the variation that can occur at the P+1 and P+3 positions, agrees well with the observed importance of the P+2 and P+5 sites for proper C-loop engagement, as well as the preponderance of prolines found at these positions in *bona fide* GSK-3 substrates (*Figure 7A,B*).

Although the second phosphorylation site in the LRP6 motif, which is three residues C-terminal to the phosphorylated Ser or Thr in the P+4 site, has been shown to be biologically important for LRP6 function, we do not observe it in either the c-motif or e-motif complex structures. Interestingly, a glutamate residue (Glu12) is present at a similar position on the pS9 auto-inhibitory peptide, which we also do not observe. It is possible that the negative charge present at this position contributes to a relatively

non-specific electrostatic interaction in a flexible region. However, previous experiments indicated that there was no increase in inhibitory activity toward GSK-3 of the doubly phosphorylated LRP6 peptide vs a peptide bearing only the first phosphorylated S/T (*Piao et al., 2008*). The second phosphorylation site may instead provide an additional site of interaction for another protein, as this region of LRP6 is accessible to solvent. One example could be an additional domain of Axin itself, as Axin has been shown to interact directly with the phosphorylated PPSP motif in the absence of GSK-3 (*MacDonald and He, 2012*; *Kim et al., 2013b*).

GSK-3 represents an important drug target for a variety of diseases, including diabetes, Alzheimer's disease, and some cancers (*Medina and Castro, 2008*; *Takahashi-Yanaga and Sasaguri, 2009*; *Wada, 2009*; *Amar et al., 2011*). Inhibiting GSK-3 in these pathways while not promoting oncogenesis through aberrant β-catenin signaling is a challenge. Many kinase inhibitors in use today are based on interactions with the nucleotide-binding region, with elaborations that target unique features of the particular enzyme. Such inhibitors would be non-selective towards the various substrates of GSK-3. Some recently developed compounds target portions of the kinase not directly involved in nucleotide binding, especially activating or inhibiting conformations of the C-helix, for example clinically approved drugs toward the Abl and EGFR tyrosine kinases (*Jura et al., 2011*). The peptide substrate-bound GSK-3 conformation observed in the crystal structures presented here suggests that it may be possible to target the three-dimensional structure formed by the rearranged C-loop in order to select for specific substrate-bound forms of GSK-3. The projection of the P+3 substrate residue inward toward a pocket formed by the rearranged C-loop (*Figure 2B*) may provide a binding surface for small molecules that interact with a particular substrate sequence at this position. Such a compound would stabilize the bound conformation and prevent turnover of a specific substrate, enabling selective targeting of one pathway controlled by GSK-3.

## Material and methods

### Constructs

GST-Axin-GBD (GSK-3 Binding Domain) fusion protein was constructed by inserting the region encoding residues 383–402 of human Axin into a modified pGEX-KG vector containing a TEV cleavage site between the GST protein and the insert. The use of EcoRI at the 5′ end for the cloning inserts the sequence GGIL between the TEV cleavage site and residue 383 of Axin.

Human LRP6 c- and e-motif–mouse GSK-3β (cDNA from ATCC, Manassas, VA) fusion proteins were constructed by replacing the first 12 residues of GSK-3β with 8 residues of the motif (PPPPTPRS for c-motif or PPPPSPCT for e-motif) by PCR and extending through residue 383 of GSK-3β. The construct also contains a non-cleavable 6xHis tag fused directly to the C-terminus of GSK-3β via PCR. This gene was then cloned into pET29b(+) using NdeI-NotI restriction sites.

A shortened form of CK1ε (residues 1–321) was constructed with a 6x-His tag fused directly to the C-terminus via PCR, and expressed in *E. coli* via the pET system in the pET21 vector (Millipore, Billerica, MA). Purification was carried out with Ni-NTA agarose beads (Qiagen, Germantown, MD) according to the manufacturer's protocol. Purified protein was stored at −20°C in 40% glycerol prior to use.

### Protein expression and purification

Expression vectors for LRP6–GSK-3 fusion proteins, GSK-3 1-383, or GSK-3 26-383 (wild-type or F67A and F93 mutants; used for kinetics experiments) were co-transformed with GST-Axin-GBD into chemically competent BL21(DE3) Codon-plus RIL cells (Stratagene, La Jolla, CA) and plated onto LB plates with 50 μg/ml kanamycin and 100 μg/ml ampicillin. A single colony was used to inoculate a 2L LB shaking culture containing the same antibiotics as listed above and grown to an approximate $OD_{595}$ of 1.0–1.2 at 37°C. Cultures were then cooled to 16°C and induced with 0.1 mM IPTG. Fresh 100 μg/ml ampicillin was added and the cultures grown for 24 hr. Cell pellets were harvested via centrifugation and stored at −80°C until purification.

Cell pellet from 4 l of culture was lysed in 40 ml of cold Lysis Buffer containing 20 mM Tris pH 7.5, 300 mM NaCl, 5% glycerol, 0.01% Triton X-100, 1 mg/ml lysozyme, 5 mg/l DNaseI (Sigma, St. Louis, MO), EDTA-free Protease Inhibitor Cocktail (Calbiochem, Billerica, MA), and 0.2 mM PMSF with an Emulsiflex homogenizer (Avestin, Toronto, Canada). Lysate was centrifuged at 39000×*g* for 1 hr at 4°C to remove insoluble material. The clarified lysate was then added to a 10 ml bed volume of Glutathione-Agarose beads (Sigma) pre-equilibrated in Lysis Buffer and run under gravity flow. The beads were then washed

with 200 ml of Wash Buffer containing 20 mM Tris pH 7.5, 300 mM NaCl, 5% glycerol, and 0.1% β-mercaptoethanol. The GSK-3/GST-Axin complex was then eluted from the beads by adding 6 × 5 ml aliquots of Wash Buffer plus 20 mM reduced L-glutathione (pH adjusted to 7.5 with NaOH). Elution fractions were analyzed by SDS-PAGE. TEV protease was added to the fractions containing the GSK-3/GST-Axin complex and allowed to incubate overnight at 4°C. Removal of the GST tag was confirmed by SDS-PAGE and samples were run over PD-10 columns pre-equilibrated with Wash Buffer according to manufacturer's protocol (GE Healthcare, Fairfield, CT) to remove glutathione at room temperature. Samples were then passed over 10 ml of glutathione-agarose beads again and the flow-through collected in order to remove cleaved GST and any uncleaved GST-Axin complex. The resulting GSK-3/Axin complex was >95% pure as analyzed by SDS-PAGE, and its concentration was approximately 0.3–0.5 mg/ml.

GSK-3β 1-383 used for the kinetics assays was produced by incubating 10 ml of the uncleaved protein with 16,000 units of lambda phosphatase (New England Biolabs, Ipswich, MA) and 2 mM $MnCl_2$ overnight at 4°C, then for 2 hr at room temperature. Removal of the lambda phosphatase, TEV cleavage, and the rest of the purification were carried out as for the AKT-phosphorylated GSK-3β 1-383. Analysis of the purified protein by western blotting showed little phosphorylation on pS9 compared to the protein before phosphatase treatment (data not shown).

## Phosphorylation and crystallization

Crystallization of the various GSK-3/Axin complexes was accomplished by dialysis. Purified GSK-3/Axin complex (12 ml) was combined with 20 mM $MgCl_2$, and 400 μM ATP (inhibitor-free structure), 30 μg of active AKT (Millipore), 10 mM $MgCl_2$, and 200 μM ATP (pS9 inhibitory peptide), or 50–100 μg of purified CK1, 10 mM $MgCl_2$, and 200 μM ATP (LRP6 c- and e-motif structures) at room temperature and injected into a 3–12 ml size Slide-a-Lyzer cassette (7000 MWCO; Thermo Scientific, Waltham, MA). The cassette was placed into a 250 ml reservoir solution of 10% PEG 35,000, 20 mM Tris pH 7.5, 300 mM NaCl, 5% glycerol, 10 mM $MgCl_2$, 200 μM ATP, and 5 mM DTT at room temperature to facilitate phosphorylation, then incubated at 4°C. Crystals formed inside the cassette after approximately 72 hr. To form the pS9–$AlF_3$ complex, the cassette was transferred to the same solution, with 200 μM ADP, 200 μM $MgCl_2$, 200 μM $Al(NO_3)_3$, and 1.2 mM NaF replacing the ATP, and incubated for 1 week at 4°C. Crystals ranging from 20 to 750 μm in diameter were harvested by excising one side of the dialysis membrane on the cassette with a razor blade and adding a cryoprotectant solution composed of 10% PEG 35,000, 20 mM Tris pH 7.5, 300 mM NaCl, 20–22% glycerol, 10 mM $MgCl_2$, 200 μM ATP (for the pS9–$AlF_3$ complex, 200 μM ADP, 200 μM $Al(NO_3)_3$, and 1.2 mM NaF were used instead of ATP) and 5 mM DTT directly to the inside of the cassette. Crystals were mounted on cryoloops (Hampton Research, Aliso Viejo, CA) and frozen directly into liquid nitrogen.

GSK-3β 1-383 phosphorylated at Ser9 (GSK-3β 1-383$^{pS9}$) used in kinase assays was produced by incubating GST-Axin/GSK-3β 1-383 with 15 μg AKT (Millipore) for approximately 48 hr in a dialysis cassette as described for crystallization, but prior to TEV cleavage. The protein solution was removed from the cassette when its volume reached ~1 ml and diluted back to 12 ml with wash buffer supplemented with 10 mM $MgCl_2$ and 200 μM ATP. The AKT was removed by applying the solution to glutathione-agarose beads. TEV cleavage and subsequent purification then proceeded as described above. Analysis of the purified protein by mass spectrometry showed that Ser9 was 24% phosphorylated.

## Data collection, analysis, and structure determination

Diffraction data were measured from crystals at 100 K at the Stanford Synchrotron Radiation Lightsource (SSRL) beamlines 12-2 or 11-1, or beamline 23ID-B of the Advanced Photon Source. Integration of images was performed with XDS (*Kabsch, 2010*), and scaling with Aimless (*Evans and Murshudov, 2013*). The crystals diffracted anisotropically, and resolution cutoffs were determined by choosing the resolution at which the half–dataset correlation $CC_{1/2}$ was approximately 0.5 and Mn(I/σ) was approximately 1.3 or better in the best direction (along the *l*-axis). The structures were solved and refined in the Phenix package (*Adams et al., 2010*). The LRP6 c- and e-motif complexes were solved by molecular replacement using PDB ID 1O9U as the starting model. The inhibitor-free, pS9 and pS9-$AlF_3$ structures were solved by rigid body refinement using the LRP6 c- and e-motif complexes. Bulk solvent parameters, TLS groups and individual temperature factors were applied throughout the refinement. Model building was carried out in Coot (*Emsley and Cowtan, 2004*). Data collection and refinement statistics are given in *Table 2*.

## GSK-3 activity and inhibition assays

GSK-3β activity was measured by incorporation of $^{32}$P into a primed, synthetic eIF2b peptide (Enzo Life Sciences, Farmingdale, NY). A typical reaction mixture contained 50 mM Tris pH 7.5, 10 mM MgCl$_2$, 0.1% β-mercaptoethanol, and 112.5 μM γ$^{32}$P-ATP at 0.02 μCi/μL (Perkin Elmer, Waltham, MA) and 224 nM GSK-3β. All measurements were performed at 30°C. The substrate peptide concentration was 20 μM or 80 μM for the inhibition assays and was varied for the determination of $K_M$ and $k_{cat}$ values. Inhibitor peptide was added at concentrations ranging from 50–1000 μM for the pS9 peptide and 5–20 μM for the LRP6 a-motif. For each measurement, a 20 μl sample was removed from the reaction and mixed immediately with 20 μl of 150 mM phosphoric acid. 35 μl of the resulting solution was then blotted onto a 96-well plate containing P81 cellulose phosphate paper and washed 5 or 10 times with 0.5% phosphoric acid. Scintillation fluid was then added, and the plate was counted in a Wallac MicroBetta 1450 liquid scintillation counter. Control reactions performed in the absence of substrate peptide showed that signal from GSK-3 autophosphorylation was not significant. Rates were calculated from the slope of the linear regression fit of $^{32}$P incorporation vs time.

$K_i$ values were calculated from initial rates by the method described in *Zhou et al. (1997)*. Briefly, an apparent $K_i$ was calculated from the equation

$$\frac{v_0}{v_i} = 1 + \frac{[I]}{K_i^{app}}$$

where $v_0$ is the rate in the absence of inhibitor, $v_i$ is the rate in the presence of the inhibitor, [I] is the concentration of the inhibitor, and $K_i^{app}$ is the apparent $K_i$. The actual $K_i$ was then calculated from $K_i^{app}$ as

$$K_i = \frac{K_i^{app}}{1 + \frac{[S]}{K_M}}$$

Individual $K_i$ values obtained from each independent experiment (n = 27 for pS9 and n = 11 for LRP6 a-motif) were averaged to obtain the reported value of $K_i$, with the error propagated from the errors in $v_0$, $v_i$, and $K_M$.

## Acknowledgements

We thank Chris Adams of the Stanford University Mass Spectrometry facility for analysis of GSK-3 Ser9 phosphorylation. JLS was supported by a Stanford Graduate Fellowship. Portions of this research were carried out at SSRL and APS beamlines supported by the US Department of Energy and the National Institutes of Health.

## Additional information

### Funding

| Funder | Grant reference number | Author |
|---|---|---|
| National Institutes of Health | GM56169 | William I Weis |
| National Institutes of Health | AG39420 | William I Weis |
| National Institutes of Health | T32 GM007276 | Michael D Enos |
| American Heart Association | Postdoctoral Fellowship | Matthew Ling-Hon Chu |
| Canadian Institutes of Health Research | Postdoctoral Fellowship | Niket Shah |

The funders had no role in study design, data collection and interpretation, or the decision to submit the work for publication.

### Author contributions

JLS, ML-HC, Conception and design, Acquisition of data, Analysis and interpretation of data, Drafting or revising the article; MDE, Conception and design, Acquisition of data, Analysis and interpretation

of data; NS, Conception and design, Analysis and interpretation of data; WIW, Conception and design, Analysis and interpretation of data, Drafting or revising the article

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
