## [Decision Letter]

Thank you for sending your work entitled “Structural basis of GSK-3 inhibition by N-terminal phosphorylation and by the Wnt receptor LRP6” for consideration at *eLife*. Your article has been favorably evaluated by a Senior editor and 3 reviewers, one of whom is a member of our Board of Reviewing Editors.

The Reviewing editor and the other reviewers discussed their comments before we reached this decision, and the Reviewing editor has assembled the following comments to help you prepare a revised submission.

The manuscript by Weis and colleagues reports several new high resolution (ca 2.1 - 2.5 A) X-ray crystal structures of the important signaling kinase GSK3. These structures are noteworthy as they contain phosphorylated inhibitory motifs from the N-terminus of GSK3 itself or derived from the Wnt signaling pathway protein LRP6. These structures reveal that the inhibitory motifs bind as predicted for primed pseudosubstrates to the enzyme in its active conformation. A special feature of the complex is the presence of an extensive interaction between the C-loop and the substrate/inhibitory motif which undergoes a significant conformational change compared to the inhibitor-free structure. These structures may also help us understand how GSK3 substrate recognition occurs and in fact how Tyr216 phosphorylation may modulate substrate specificity. This study gives insight into how Tyr216 phosphorylation can modestly activate the kinase. The conserved nature of GSK3 Phe93, from the C-loop, is now explained and tested here by mutagenesis, showing its importance. A role of a conserved Arg in GSK3 interacting with the phospho-site of the primed substrate is suggested. Some technically impressive aspects of the study include the use of chimeric LRP6-GSK3 to capture the inhibited conformation and the use of kinases to obtain cleanly phosphorylated GSK3 for crystallization, and the use of ALF3- to visualize the “transition state”. Overall, this is a well-done and significant manuscript that should be of broad interest to *eLife* readers.

Principal concerns to be addressed:

1) There were several concerns raised about the kinetic measurements which should be addressed. The k_cat_ values, in the range of 0.04 s^-1^ (Table 1, Figure 1) are on the low side compared to many other active kinases (e.g., PKA is about 20 s^-1^). It would be helpful if the authors could compare their rate constants with any other GSK3 rate constants recorded in the literature to see if this has been found in other labs. The background phosphate values (time zero) are quite high at low inhibitor peptide as shown in Figure 1 and get lower with increasing inhibitor peptide additions. It would be helpful if the authors could comment on why this might be the case and also indicate how they dealt with this in calculating rates. For example, were the rates based on slopes (probably best) or were they product/time without background subtraction? Why such large error bars for the K_M_ determinations? It doesn't look like the authors used a standard model for substrate competitive inhibition to fit their data for inhibition by the pS9 and LRP6 peptides. Was non-linear regression used? What software? What inhibition model?

2) It does not appear that the catalytic activity of the phospho-GSK3 enzymes were measured (or at least they are not reported). These would presumably show reduced activity compared to the non-N-terminally phosphorylated enzyme but it would be nice to demonstrate this and report the quantitative data. If the activity of these forms is below the level of detection, that could be stated (and the detection threshold mentioned).

Additional comments for consideration:

- While it certainly seems reasonable that the present work provides insight regarding recognition of substrate peptides (and perhaps also of primed substrate peptides), it may be excessive to claim that the structures presented here reveal the basis of primed substrate recognition. These peptides are inhibitors, not substrates, and indeed it seems that none of the structures reveal an ordered residue in the position expected for a phosphoacceptor. Further, intermingled discussion of the peptides as inhibitors and substrates is potentially confusing to the reader. Our advice would be to discuss implications for substrate recognition in a separate section, with clear acknowledgement of the limitations of the interpretation.

- The authors refer to the phosphorylated residue as P+4. This is understandable, given the P+4 priming phenomenon. However, is this a structurally sound interpretation? In the superposition with PKA bound to PKI, does the “P+1” in the peptides studied here superimpose with the P+1 in PKI? Are there backbone hydrogen bonds between peptide and activation loop in common between the structures described here and other kinase/peptide substrate complexes that allow the authors to define the register and conclude that the phosphorylated Ser/Thr is indeed in a P+4 position.

- Were the C-loop mutants studied in the context of inhibition by LRP6 or pSer9?

- The authors state that this is the first kinase complexed to a primed substrate. While this may be true for Ser/Thr kinases, the authors should cite the following, “Structural recognition of an optimized substrate for the ephrin family of receptor tyrosine kinases” (Davis et al, FEBS J. 2009 Aug;276(16):4395-404), as a structure that shows a pTyr primed substrate complexed to an ephrin receptor kinase.

- The authors state that, “Most kinase inhibitors in use today stabilize the inactive conformations...” While lapatinib, sorafenib, and imatinib are examples of this, erlotinib, gefitinib, dasatinib, stutent, and vemurafinib bind to active conformations of kinases. Thus, we would suggest modifying this point.

- Is it clear from the structures that an N-tail from one enzyme is not extending into the active site of a neighbor in the crystal lattice (ie intermolecular interaction)? The authors could comment on this. If there is any doubt in solution, they could measure the rate of GSK3 kinase activity at different phospho-GSK3 concentrations.

- The authors state in their Abstract and ultimate sentence that their results will aid design of GSK-3β inhibitors that are selective for specific pathways, but it is not obvious how. The authors may wish to better flesh out this claim or consider omitting it.

---

## [Author Response]

*Principal concerns to be addressed*:

*1) There were several concerns raised about the kinetic measurements which should be addressed. The k*_*cat*_
*values, in the range of 0.04 s*^*-1*^
*(*Table 1*,*
Figure 1*) are on the low side compared to many other active kinases (e.g., PKA is about 20 s*^*-1*^*). It would be helpful if the authors could compare their rate constants with any other GSK3 rate constants recorded in the literature to see if this has been found in other labs*.

Note: Some small errors in data analysis were corrected in the revised manuscript. The final values of kinetic constants have changed slightly, but the conclusions are unaffected. In particular, the LRP6 a-motif peptide inhibits GSK-3 approximately 40x more strongly than the S9 peptide, rather than 16x reported in the previous version. The other notable error was in conversion of rates from pmol/sec to 1/sec, which resulted in k_cat_ values (and thus k_cat_/K_M_ values) smaller by a factor of two.

The activity is indeed considerably lower than PKA and some other kinases. There have been very few measurements of rate constants reported for GSK-3, as the vast majority of studies use simple qualitative endpoint assays, for example western blotting to check for formation of phosphorylated substrates, to measure activity. Also, as we note in the text, direct comparison with other GSK-3 studies is not straightforward because different groups have used different substrate sequences, most of which are processive (i.e., multiple phosphorylations at successive *n-4* sites). Nonetheless, we can compare values from the handful of studies that report kinetic constants in the table below.

Fiol et al (1987 and 1990) [1, 2] Wang et al (1994) [3] and Dajani et al (2003) [4] all give K_M_ values for other GSK-3β substrate peptides that agree with our results to within an order of magnitude (and are often larger than ours). Fiol et al (1987) also gives a k_cat_ for what the authors later determined is rabbit GSK-3α (Fiol et al, 1990) that is more than an order of magnitude lower than ours (approximately 0.0014 s^-1^ as calculated from their data). Dajani et al. also reported a k_cat_ of 0.7, but no units were provided. The assay referenced in that paper uses rate units of pmol min^-1^, so if we assume that the k_cat_ is reported in units of min^-1^ then the value is 0.012 s^-1^, somewhat smaller than our value of 0.079 s^-1^.ReferenceSubstratek_cat_K_M_ (µM)Fiol et al, 1987Primed peptide based on glycogen synthase0.0014 sec^−1^2Fiol et al, 1987Primed peptide based on glycogen synthasen.d.2Fiol et al, 1987Primed peptide based on glycogen synthase with second GSK-3 target serine mutated to alaninen.d.6Fiol et al, 1987Primed peptide based on glycogen synthase with thirdGSK-3 target serine mutated to alaninen.d.3Fiol et al, 1987Primed peptide based on glycogen synthase with fourth GSK-3 target serine mutated to alaninen.d.3Wang et al, 1994Phospho-CREB peptiden.d.200Wang et al, 1994Myelin basic proteinn.d.59Wang et al, 1994κ-Caseinn.d.114Wang et al, 1994Phosvitinn.d.200Wang et al, 1994Phosphatase inhibitor-2?16Dajani et al, 2001Primed peptide based on glycogen synthase??70

References:

1. Fiol, C.J., Mahrenholz, A.M., Wang, Y., Roeske, R.W., and Roach, P.J. (1987). Formation of protein kinase recognition sites by covalent modification of the substrate. Molecular mechanism for the synergistic action of casein kinase II and glycogen synthase kinase 3. J Biol Chem *262,* 14042-14048.

2. Fiol, C.J.; Wang, A.; Roeske, R.W.; and Roach, P.J. (1990). Ordered Multisite Protein Phosphorylation. Analysis of glycogen synthase kinase 3 action using model peptide substrates. J Biol Chem *265,* 6061-6065.

3. Wang, Q.M., Fiol, C.J., DePaoli-Roach, A.A., and Roach, P.J. (1994). Glycogen synthase kinase-3 beta is a dual specificity kinase differentially regulated by tyrosine and serine/threonine phosphorylation. J Biol Chem *269,* 14566-14574.

4. Dajani, R., Fraser, E., Roe, S.M., Yeo, M., Good, V.M., Thompson, V., Dale, T.C., and Pearl, L.H. (2003). Structural basis for recruitment of glycogen synthase kinase 3β to the axin-APC scaffold complex. EMBO J *22,* 494-501.

*The background phosphate values (time zero) are quite high at low inhibitor peptide as shown in*
Figure 1
*and get lower with increasing inhibitor peptide additions. It would be helpful if the authors could comment on why this might be the case and also indicate how they dealt with this in calculating rates. For example, were the rates based on slopes (probably best) or were they product/time without background subtraction? Why such large error bars for the K*_*M*_
*determinations? It doesn't look like the authors used a standard model for substrate competitive inhibition to fit their data for inhibition by the pS9 and LRP6 peptides. Was non-linear regression used? What software? What inhibition model*?

Rates were calculated as the slopes of ^32^P incorporated into the substrate peptide versus time. Unfortunately the solid black symbols for the inhibitor free data in the original Figure 1 were hard to read: although linear regression for the inhibitor-free data gives a non-zero y intercept at t = 0, the actual data points for t = 0 (now shown as clear circles for the inhibitor free data) are in fact comparable to the other data. This likely corresponds to the transition between a burst phase and steady-state kinetic regime. In these experiments, ATP and inhibitor peptides (or control with no inhibitor) were preincubated with enzyme prior to addition of the substrate peptide. The rapid burst in the absence of inhibitor peptide can be attributed to rapid binding of substrate peptide, which reduces the time needed to form the ES complex. In the presence of preincubated inhibitor peptide, however, formation of the ES complex requires dissociation of the inhibitor peptide from the enzyme, so an initial fast turnover does not occur. Although this difference may ultimately give some interesting insights into the actual mechanism of the enzyme, the point of this figure is to compare the relative inhibitory potencies of the S9 and LRP6 a-motif peptides.

There are two principal reasons for the large error bars in the K_M_ measurements. First, filter binding is itself a somewhat noisy assay, and individual rates were not always perfectly consistent across different days. Second, the inherent noise in the assay was compounded for the mutants by the low signal.

As described in the Materials and methods section, the inhibition constants were obtained using the method in the [59] reference.

*2) It does not appear that the catalytic activity of the phospho-GSK3 enzymes were measured (or at least they are not reported). These would presumably show reduced activity compared to the non-N-terminally phosphorylated enzyme but it would be nice to demonstrate this and report the quantitative data. If the activity of these forms is below the level of detection, that could be stated (and the detection threshold mentioned)*.

This proves to be an extremely difficult experiment, which is probably why it has not been reported previously despite 20 years of knowledge that phosphorylation of Ser9 inhibits GSK-3β activity. The reviewers reasonably assumed that this would be straightforward given the crystal structures, but in fact is very difficult to obtain 100% phosphorylated material. We attempted in vitro phosphorylation using AKT for native GSK-3, or CK1 for the LRP6 chimeras, but we obtained only a few percent phosphate incorporation (measured by mass spectrometry). As GSK-3 is not efficiently phosphorylated, the extreme concentration of GSK-3 along with AKT (or CK1 for the LRP6 chimeras) produced by dialysis against high molecular weight PEG enabled sufficient production of the phosphorylated enzyme for crystallization. Note that although the peptide is phosphorylated in the structure, we did not know what fraction of the total enzyme input to crystallization was phosphorylated.

To address the reviewers’ request, we used the crystallization protocol to produce phosphorylated full-length GSK-3β, but halted the dialysis prior to crystallization. Mass spectrometry showed that this material is 24 % phosphorylated. The partially phosphorylated material shows poor activity relative to the non-phosphorylated enzyme (produced by phosphatase treatment of the full-length enzyme, or the truncated version lacking the autoinhibitory N-terminus). Although we cannot produce fully phosphorylated material, the experiment clearly shows that the in vitro phosphorylated material is inhibited. This is now described in the text and shown in Figure 1 and Table 1.

Given the clear data reported here and earlier on inhibition by synthetic phosphorylated peptides in trans, we do not feel that it would be worth the considerable costs in time and money needed to repeat this experiment with the LRP6 chimeras.

*Additional comments for consideration*:

*- While it certainly seems reasonable that the present work provides insight regarding recognition of substrate peptides (and perhaps also of primed substrate peptides), it may be excessive to claim that the structures presented here reveal the basis of primed substrate recognition. These peptides are inhibitors, not substrates, and indeed it seems that none of the structures reveal an ordered residue in the position expected for a phosphoacceptor. Further, intermingled discussion of the peptides as inhibitors and substrates is potentially confusing to the reader. Our advice would be to discuss implications for substrate recognition in a separate section, with clear acknowledgement of the limitations of the interpretation*.

We agree that the distinction of inhibitor and true substrate peptides was muddled in the original manuscript. The point was that comparison with a known active kinase like Thr197-phosphorylated PKA shows that the enzyme itself is in an active conformation, indicating that the inhibitors work as pseudo-substrates. We have carefully maintained this distinction in the revised paper. Given these structural data, as well as the comparison with known true substrate sequences (see next point), it is very likely that the binding mode visualized in these structures represents the mode of binding of true primed substrates. Moreover, the effect of F93 mutation clearly shows that the interaction with the C-loop is relevant to the action of the kinase on a true substrate.

*- The authors refer to the phosphorylated residue as P+4. This is understandable, given the P+4 priming phenomenon. However, is this a structurally sound interpretation? In the superposition with PKA bound to PKI, does the “P+1” in the peptides studied here superimpose with the P+1 in PKI? Are there backbone hydrogen bonds between peptide and activation loop in common between the structures described here and other kinase/peptide substrate complexes that allow the authors to define the register and conclude that the phosphorylated Ser/Thr is indeed in a P+4 position*.

We are not entirely sure about the reviewers’ question as to whether this is a “structurally sound interpretation”. The P+4 assignment is based on the known properties of GSK-3 substrates. The electron density of the peptide is well defined at this resolution, so there is no ambiguity regarding the direction of the polypeptide nor the visible side chain assignments. We have now added a comparison to PKA:PKI and other kinases bound to substrates, showing that the orientation and structures of these other kinases in the P+1 site is similar (text and Figure 5). This further supports the notion that the binding mode of the inhibitory peptides corresponds closely to that of true primed substrates.

*- Were the C-loop mutants studied in the context of inhibition by LRP6 or pSer9*?

No. The F93 mutation was made to show that this residue interacts with real substrates as part of catalysis, supporting the claim that the structures visualize an active conformation and that the inhibitors are pseudo substrates. Given the poor activity of 5 the F93 mutants, it is doubtful that we would be able to detect further inhibition by the peptides.

*- The authors state that this is the first kinase complexed to a primed substrate. While this may be true for Ser/Thr kinases, the authors should cite the following, “Structural recognition of an optimized substrate for the ephrin family of receptor tyrosine kinases” (Davis et al, FEBS J. 2009 Aug;276(16):4395-404), as a structure that shows a pTyr primed substrate complexed to an ephrin receptor kinase*.

Thank you for this suggestion; we have now cited this paper in the Introduction.

*- The authors state that, “Most kinase inhibitors in use today stabilize the inactive conformations...” While lapatinib, sorafenib, and imatinib are examples of this, erlotinib, gefitinib, dasatinib, stutent, and vemurafinib bind to active conformations of kinases. Thus, we would suggest modifying this point*.

Thank you, we have clarified this point in the last paragraph.

*- Is it clear from the structures that an N-tail from one enzyme is not extending into the active site of a neighbour in the crystal lattice (i.e., intermolecular interaction)? The authors could comment on this. If there is any doubt in solution, they could measure the rate of GSK3 kinase activity at different phospho-GSK3 concentrations*.

Addressing whether the S9 interaction is truly autoinhibitory versus trans-inhibitory is an interesting question, but this was not our focus as we are interested in comparing the interaction of the N-terminal peptide (presumed to be cis) with the LRP6 peptides, which are of course part of a different molecule in vivo. To answer the reviewer’s question, we cannot rule out that the N-tail of one molecule is binding in the active site of another in the crystals, as the connection is disordered and the arrangement of molecules in the lattice might allow this to happen. We are also somewhat restricted in the range of concentrations to test GSK-3 activity due to the signal-to-noise of the assay at low concentrations and the need to keep GSK-3 soluble and to maintain steady-state conditions at high concentrations. We have noted in the Discussion that there is a “presumed” kinetic advantage of having the inhibitory peptide in cis in GSK-3.

*- The authors state in their Abstract and ultimate sentence that their results will aid design of GSK-3β inhibitors that are selective for specific pathways, but it is not obvious how. The authors may wish to better flesh out this claim or consider omitting it*.

We have clarified this proposal in the last paragraph.